# Model-based whole-brain perturbational landscape of neurodegenerative diseases

Yonatan Sanz Perl[1,2,3,4]*, Sol Fittipaldi[2,3], Cecilia Gonzalez Campo[2,3], Sebastián Moguilner[5,6], Josephine Cruzat[4,6], Matias E Fraile-Vazquez[3], Rubén Herzog[6], Morten L Kringelbach[7,8,9,10], Gustavo Deco[4,11,12,13,14], Pavel Prado[6,15], Agustin Ibanez[2,3,5,6,16], Enzo Tagliazucchi[1,2,3,6]

[1]Department of Physics, University of Buenos Aires, Buenos Aires, Argentina; [2]National Scientific and Technical Research Council (CONICET), CABA, Buenos Aires, Argentina; [3]Cognitive Neuroscience Center (CNC), Universidad de San Andrés, Buenos Aires, Argentina; [4]Center for Brain and Cognition, Computational Neuroscience Group, Universitat Pompeu Fabra, Barcelona, Spain; [5]Global Brain Health Institute, University of California, San Francisco, San Francisco, United States; [6]Latin American Brain Health Institute (BrainLat), Universidad Adolfo Ibáñez, Santiago, Chile; [7]Department of Psychiatry, University of Oxford, Oxford, United Kingdom; [8]Center for Music in the Brain, Department of Clinical Medicine, Aarhus University, Århus, Denmark; [9]Life and Health Sciences Research Institute (ICVS), School of Medicine, University of Minho, Braga, Portugal; [10]Centre for Eudaimonia and Human Flourishing, University of Oxford, Oxford, United Kingdom; [11]Department of Information and Communication Technologies, Universitat Pompeu Fabra, Barcelona, Spain; [12]Institució Catalana de la Recerca i Estudis Avancats (ICREA), Barcelona, Spain; [13]Department of Neuropsychology, Max Planck Institute for Human Cognitive and Brain Sciences, Leipzig, Germany; [14]School of Psychological Sciences, Monash University, Clayton, Australia; [15]Escuela de Fonoaudiología, Facultad de Odontología y Ciencias de la Rehabilitación, Universidad San Sebastián, Santiago, Chile; [16]Trinity College Institute of Neuroscience (TCIN), Trinity College Dublin, Dublin, Ireland

*For correspondence:
yonatan.sanz@upf.edu

**Competing interest:** The authors declare that no competing interests exist.

**Abstract** The treatment of neurodegenerative diseases is hindered by lack of interventions capable of steering multimodal whole-brain dynamics towards patterns indicative of preserved brain health. To address this problem, we combined deep learning with a model capable of reproducing whole-brain functional connectivity in patients diagnosed with Alzheimer's disease (AD) and behavioral variant frontotemporal dementia (bvFTD). These models included disease-specific atrophy maps as priors to modulate local parameters, revealing increased stability of hippocampal and insular dynamics as signatures of brain atrophy in AD and bvFTD, respectively. Using variational autoencoders, we visualized different pathologies and their severity as the evolution of trajectories in a low-dimensional latent space. Finally, we perturbed the model to reveal key AD- and bvFTD-specific regions to induce transitions from pathological to healthy brain states. Overall, we obtained novel insights on disease progression and control by means of external stimulation, while identifying dynamical mechanisms that underlie functional alterations in neurodegeneration.

## Editor's evaluation

Sanz Perl and colleagues provide important insights regarding the application of computational brain models from neurodegenerative diseases to evaluate brain stimulation protocols in silico. Solid

evidence is provided for the disease-specificity of the framework, however, the real-world impact of such stimulation protocols to alleviate psychiatric and neurological symptoms remains to be evaluated.

## Introduction

Neurodegenerative diseases and dementia represent an increasing social and economic burden to worldwide health (*GBD 2019 Dementia Forecasting Collaborators, 2022*), with rising prevalence and incidence in countries with an aging population and specially across underrepresented populations from the developing world, where classical biomarkers and treatments are not yet massively available (*Hou et al., 2019*; *Mukadam et al., 2019*; *Parra et al., 2018*; *Parra et al., 2023*). Alzheimer's disease (AD) and behavioral variant frontotemporal dementia (bvFTD) are among the most prevalent neurodegenerative diseases, each linked to specific pathophysiology and to highly heterogeneous and atypical manifestation, hindering detection and diagnosis (*Musa et al., 2020*; *Parra et al., 2018*). Neuroimaging methods such as positron emission tomography (PET) (*Engler et al., 2008*; *Foster et al., 2007*; *Jack et al., 2018*; *Mosconi et al., 2008*; *Nordberg, 2004*), functional magnetic resonance imaging (fMRI) (*Hafkemeijer et al., 2015*; *Jalilianhasanpour et al., 2019*; *Moguilner et al., 2021*), and electroencephalography (EEG) *Dottori et al., 2017*; *Lindau et al., 2003*; *Nishida et al., 2013*; *Nishida et al., 2011*; *Cruzat et al., 2023* have been employed extensively to profile these diseases based on their underlying whole-brain activity patterns, and to develop automated classifiers capable of assisting clinical decision-making (*Kim et al., 2019*; *Moguilner et al., 2021*; *Herzog et al., 2022*). In spite of these advances, the limited knowledge of the large-scale network mechanisms of neurodegenerative dementias hinders the development of interventions oriented to improve brain health and subjective well-being. Moreover, without addressing the underlying neurobiological mechanisms, purely data-driven metrics can be heterogeneous and difficult to interpret, in particular when obtained from large-scale studies which are capable of picking up small differences that arise due to confounding factors (*Deco and Kringelbach, 2014*).

Several methods capable of externally modulating brain activity have been proposed to treat AD and bvFTD, including transcranial magnetic stimulation (TMS) (*Antczak et al., 2018*; *Nardone et al., 2012*), direct and alternating electrical current stimulation (tDCS and tACS) (*Benussi et al., 2021*; *Benussi et al., 2020a*; *Bréchet et al., 2021*; *Ferrucci et al., 2008*; *Prehn and Flöel, 2015*; *Sprugnoli et al., 2021*; *Zhou et al., 2022*; *Pini et al., 2022*; *Birba et al., 2017*), and ultrasound pulse stimulation (*Liu et al., 2021*), among others. Overall, the use of transcranial stimulation techniques resulted in mixed results, with only some of these studies reporting promising results in terms of the restoration of disease-specific cognitive impairments and functional abnormalities. A frequent limitation is given by their exploratory nature, highlighting the need for principled methods to determine the brain regions to be stimulated, as well as relevant parameters such as the scalp location of the stimulation devices (montage), and the intensity and frequency of the delivered perturbation. These limitations could be addressed using whole-brain models of brain activity that incorporate multiple sources of experimental data to reproduce realistic estimates of brain dynamics and functional connectivity (*Menardi et al., 2022*; *Deco et al., 2019*; *Wang et al., 2015*). After parameter optimization, models can be used as a sandbox to test the outcome of different forms of external perturbation (*Cofré et al., 2020*). Previous research supports the feasibility of fitting whole-brain phenomenological models to fMRI data acquired from AD patients, suggesting mechanistic explanations for reduced whole-brain synchronization and small-worldness of functional networks in the patients (*Demirtaş et al., 2017*). However, the use of whole-brain models to infer and reverse the large-scale network mechanisms underlying neurodegenerative dementias (AD and bvFTD) remains largely unexplored.

Computational models of brain activity present a trade-off between interpretability and accuracy to represent empirical data, the latter being closely related to the number of parameters that are optimized during the fitting procedure (*Cofré et al., 2020*; *Ipina et al., 2020*). Increasing the size of parameter space is necessary to model brain states characterized by spatially heterogeneous changes in neural dynamics, as is the case of neurodegenerative disorders, where atrophy has been linked to alterations in the local excitation/inhibition (E/I) balance (*Mehta et al., 2013*; *Lopatina et al., 2019*; *Maestú et al., 2021*; *Palop and Mucke, 2016*). Neurodegenerative diseases such AD and bvFTD are characterized by heterogeneity across different brain levels involving both structure and dynamics

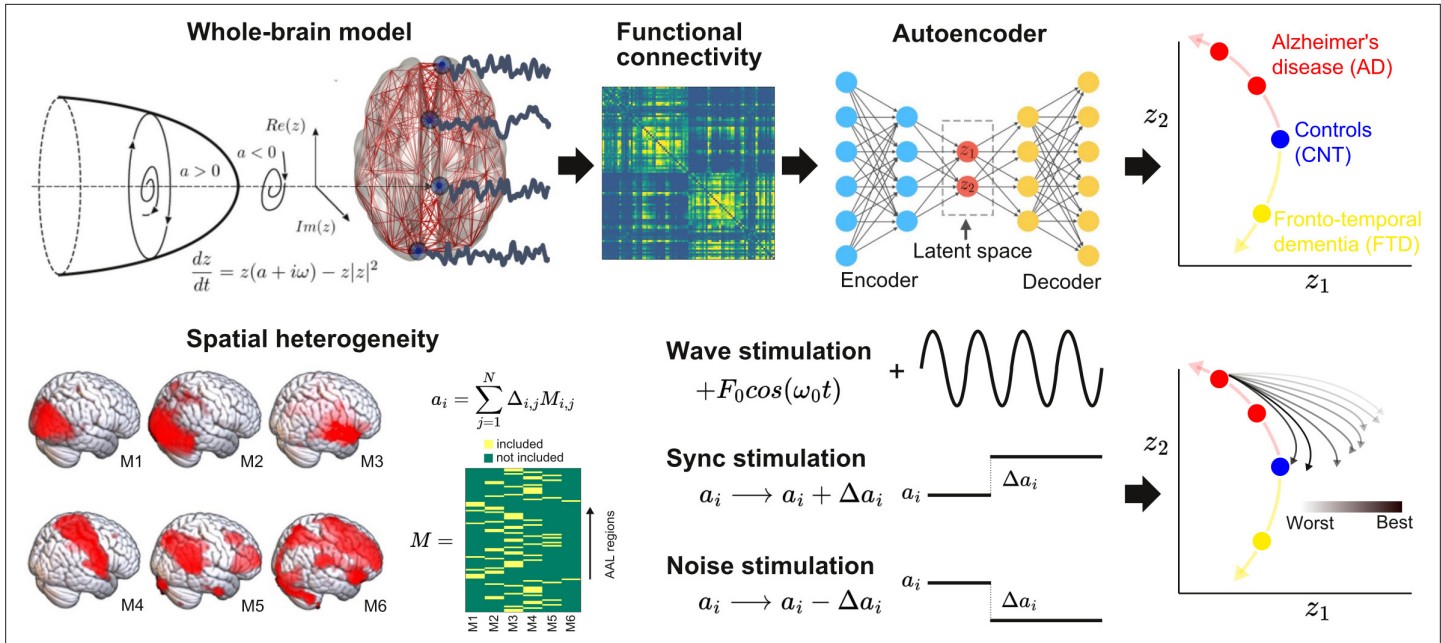

**Figure 1.** Methodological outline. A phenomenological whole-brain model (normal mode of a Hopf bifurcation) was implemented at nodes defined by the Automated Anatomical Labeling (AAL) parcellation and coupled with the anatomical connectome. Different priors were explored to induce spatial heterogeneity in the model (i.e., variation in the local bifurcation parameters). The model was tuned to reproduce the empirical functional connectivity (FC) for each condition (CNT, Alzheimer's disease [AD], behavioral variant frontotemporal dementia [bvFTD]), and the resulting parameters were represented in a latent space using a variational autoencoder, facilitating comparison between groups. Finally, the three different perturbations (Wave, Sync, Noise) were introduced in the model, resulting in a set of trajectories in latent space (one per pair of homotopic AAL regions).

(**Mehta et al., 2013**; **Peet et al., 2021**; **Seelaar et al., 2011**; **Verdi et al., 2021**; **Prado et al., 2023**), challenging the assumptions of spatially homogeneous biomarker models. Moreover, the issue of interpretability manifests when the impact of simulated stimulation is considered. In which ways whole-brain activity can be modified? How do these changes depend on the selected stimulation site? What is the relationship between the perturbed dynamics and those observed in target healthy brain states? Providing answers to these questions (and thus exploring the landscape of perturbation-induced states, i.e., the *perturbational landscape*) becomes increasingly difficult as the number of model parameters is increased.

We implemented a two-stage procedure to tackle these issues by means of whole-brain activity models. First, we introduced empirical structural connectivity data to couple the local dynamics, as well as priors to modulate local model parameters based on atrophy maps of different neurodegenerative diseases, such as AD, bvFTD, and Parkison's disease (PD), with the hypothesis that disease-specific maps are capable of improving the model fit to the fMRI data of the corresponding disease. Next, we fitted these models to the empirical data, which allowed us to identify potential mechanisms underlying the changes observed in AD and bvFTD, relative to the relationship between the dynamical regime of local activity and whole-brain connectivity patterns. Finally, we used variational autoencoders (VAE) (**Perl et al., 2020**) to obtain a low-dimensional representation of brain activity, investigating how positioning in this latent space is related to the diagnosis and the severity of the disease. Our analysis concluded with the systematic exploration of different stimulation protocols, visualizing the outcome as trajectories in latent space, and thus allowing us to interpret the effects of the stimulation in terms of their capacity to restore healthy whole-brain activity patterns.

## Results

The outline of the procedure followed in this work is shown in **Figure 1**. First, we implemented a whole-brain model with local dynamics (one per node in the Automated Anatomical Labeling atlas [AAL]) (**Tzourio-Mazoyer et al., 2002**) given by the normal form of a Hopf bifurcation (**Deco et al., 2017**). Depending on the bifurcation parameter a (related to the excitation/inhibition ratio) these dynamics

present two qualitatively different behaviors: fixed-point dynamics (a < 0) and oscillations around a limit cycle (a > 0). When noise is added to the model, dynamics close to the bifurcation (a ≈ 0) change stochastically between both regimes, giving rise to oscillations with complex amplitude modulations. Regions were coupled by the anatomical connectivity matrix obtained from diffusion tensor imaging (DTI) measurements. The model was used to simulate phenomenological time series (i.e., the output of the model is interpreted as the measured BOLD signal, without the need to apply biophysical transformations). Then, the whole-brain functional connectivity (FC) matrices computed from the simulated time series by means of Pearson's correlation were encoded into a two-dimensional space using a deep learning architecture known as VAE (*Perl et al., 2020*).

Briefly, VAEs are deep neural networks with autoencoder (AE) architecture (*Kingma and Welling, 2013*), which are trained to map inputs to probability distributions in latent space by minimizing the error between the input and the output. This output corresponds to the input as reproduced from the latent space values. Moreover, VAEs can be regularized during the training process to produce meaningful outputs after the decoding step, as well as to ensure continuity between the outputs and the corresponding choice of latent space values. The most common architecture of this network can be subdivided into three parts: the encoder network, the middle variational layer (with units corresponding to the latent space), and decoder network. The encoder transforms the input into the latent space, which is typically of much lower dimension than the input and output layers. On the other hand, the decoder converts the values of the units in the latent space to the output space.

The whole-brain model had different bifurcation parameters for each AAL region, constrained by the spatial maps of the following anatomical priors: resting state networks (RSN), regions with different brain atrophy values, random assignment of regions to the spatial maps, and maps corresponding to an equipartition of neighboring regions. For each region in the prior, an independent parameter (Δ) was fitted, then these parameters were combined to obtain the local bifurcation parameters of the model (note that regions were potentially overlapping) (*Ipina et al., 2020*). Following *Ipina et al., 2020*, these free parameters were optimized using a genetic algorithm to optimize the agreement with fMRI FC from an elderly healthy control (CNT), AD, and bvFTD patients. We explored different forms of in silico external perturbations that emulate specific empirical perturbation protocols. Wave stimulation (periodic perturbation delivered at the dominant frequency of the local BOLD fluctuations)

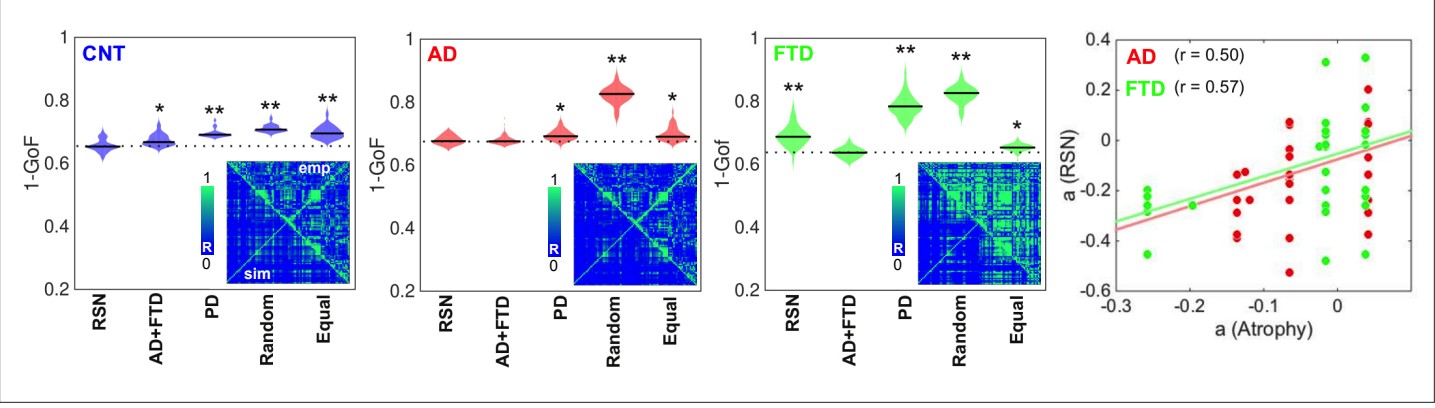

**Figure 2.** Fitting the whole-brain model to the empirical data. The violin plots display 1 – GoF values (300 independent realizations of parameter fitting) for CNT, Alzheimer's disease (AD), and behavioral variant frontotemporal dementia (bvFTD) using anatomical priors based on resting state networks (RSN), AD, and bvFTD atrophy maps (AD + FTD), Parkinson's disease atrophy map (PD), random assignment (Random), and equally sized groups of nodes defined by anatomical proximity (Equal) (* and ** indicate large [|d| >0.8] and very large [|d| >1.3] effect sizes according to Cohen's d, computed in each case against the best fitting prior; e.g., in the models fitted to the CNT and AD groups, the effect size was computed against the RSN and AD + FTD priors, respectively). Insets present the empirical ('emp,' below diagonal) and the best simulated ('sim,' above diagonal) FC matrices, with contradiagonal matrix entries added for visualization purposes. The rightmost panel shows the bifurcation parameters obtained using the RSN vs. AD + FTD atrophy priors for AD (red) and bvFTD (green), together with the corresponding Pearson correlation values and least-squares linear fits.

The online version of this article includes the following figure supplement(s) for figure 2:

**Figure supplement 1.** Atrophy maps.

**Figure supplement 2.** Correlations between disease-atrophy maps.

**Figure supplement 3.** Fitting the whole-brain model to the empirical data.

imitates the characteristics of tACS, considering that both approaches are based on an external periodical driver applied to the brain. The specific application of the nodal natural oscillatory frequency is based on reports that suggest electrophysiological oscillations can be synchronized by in-phase tACS stimulation (*Helfrich et al., 2014*), even though this mechanism has been recently disputed (*Lafon et al., 2017*) and further research is required for its validation. On the other hand, the simulated Sync/Noise stimulation increases/decreases the overall value of the bifurcation parameter underlying the switching of the dynamical regime of a specific brain region. This can be associated with a direct alteration in nodal neural excitability, which resembles the impact created by tDCS stimulation (*Nitsche and Paulus, 2000*). After systematically applying these perturbations to all pairs of homotopic nodes and encoding the resulting FC matrices, we obtained low-dimensional perturbational landscapes, consisting of trajectories in latent space parametrized by the stimulation intensity. In turn, these trajectories were classified depending on how closely they brought the dynamics to a predefined target state, in this case, that of the healthy control group.

## Fitting the whole-brain model

The results of model fitting are shown in *Figure 2*. The violin plots display 1 – goodness of fit (GoF) values for CNT, AD, and bvFTD using the following anatomical priors: RSN (*Beckmann and Smith, 2004*), AD, and bvFTD atrophy maps (AD + FTD), Parkinson's disease atrophy map (PD), random assignment (Random), and equally sized groups of nodes defined by anatomical proximity (Equal). Note that we combined the anatomical priors (i.e., atrophy maps) from AD and FTD due to their high level of spatial correlation ($R = 0.75$, $p<0.0001$, see *Figure 2—figure supplements 1 and 2*). These values were obtained for 300 independent realizations of parameter optimization with a genetic algorithm (*Ipina et al., 2020*). Here, the use of maps from a different neurodegenerative disease (PD, without dementia and less characterized by atrophy) was implemented as an additional control to compare the resulting GoF with the values obtained using disease-specific atrophy maps. We found that the best fits were obtained using RSN for CNT, AD + FTD atrophy maps or RSN for AD, and AD + FTD atrophy maps for bvFTD (in all cases with $1 - \text{GoF} \approx 0.6$, similar to values reported in previous publications based on different datasets) (*Deco et al., 2017*; *Demirtaş et al., 2017*; *Sanz Perl et al., 2021*), showing that atrophy maps contain meaningful spatial heterogeneities that improve the fit to the empirical whole-brain FC matrix. We repeated the same procedure but considering separately the AD and FTD maps, and evaluated the goodness of fit for each condition, as shown in supplementary material *Figure 2—figure supplement 3*. We found that the best fits were obtained using RSN for CNT, RSN, AD, or FTD atrophy maps for AD, and the FTD atrophy map for FTD (with $1 - \text{GoF} \approx 0.6$) (see supplementary material).

The insets show the empirical ('emp,' below diagonal) and simulated ('sim,' above diagonal) FC matrices. While good correspondence between empirical and simulated data can be observed in the off-diagonal blocks of the FC matrix, a comparatively lower homotopic FC was obtained for the simulated dynamics, as can be expected from known limitations of the anatomical connectome (*Reveley et al., 2015*). Due to the use of SSIM as a target optimization function (*Ipina et al., 2020*), the optimal simulated matrices captured both the average FC and the relative differences between matrix entries, which jointly minimize the Euclidean and correlation distances, respectively.

Finally, to facilitate further comparison, we used the same prior (RSN) to fit all the groups. The panels in *Figure 2* show that the GoF obtained using the RSN prior was similar to that obtained using disease-specific atrophy maps. Moreover, after fitting the model and converting the $\Delta$ to bifurcation parameters, we correlated the regional parameters obtained for the RSN and AD + FTD atrophy maps. Even though the RSN prior resulted in significantly lower GoF for the bvFTD group, we found an acceptable level of similarity ($R \approx 0.5$) between the local bifurcation parameters obtained using the RSN and AD + FTD atrophy map priors. Based on this result, in the rest of this work we adopted the necessary approximation of using the RSN parcellation to constrain local bifurcation parameters, in order to obtain models based on the same prior that could be compared across the different pathologies.

## Comparison of model parameters between groups

Here and in the following analysis, we divided the AD group into two subgroups of different severity (AD- and AD+), as determined using non-atrophy measures of brain disease obtained from the patients.

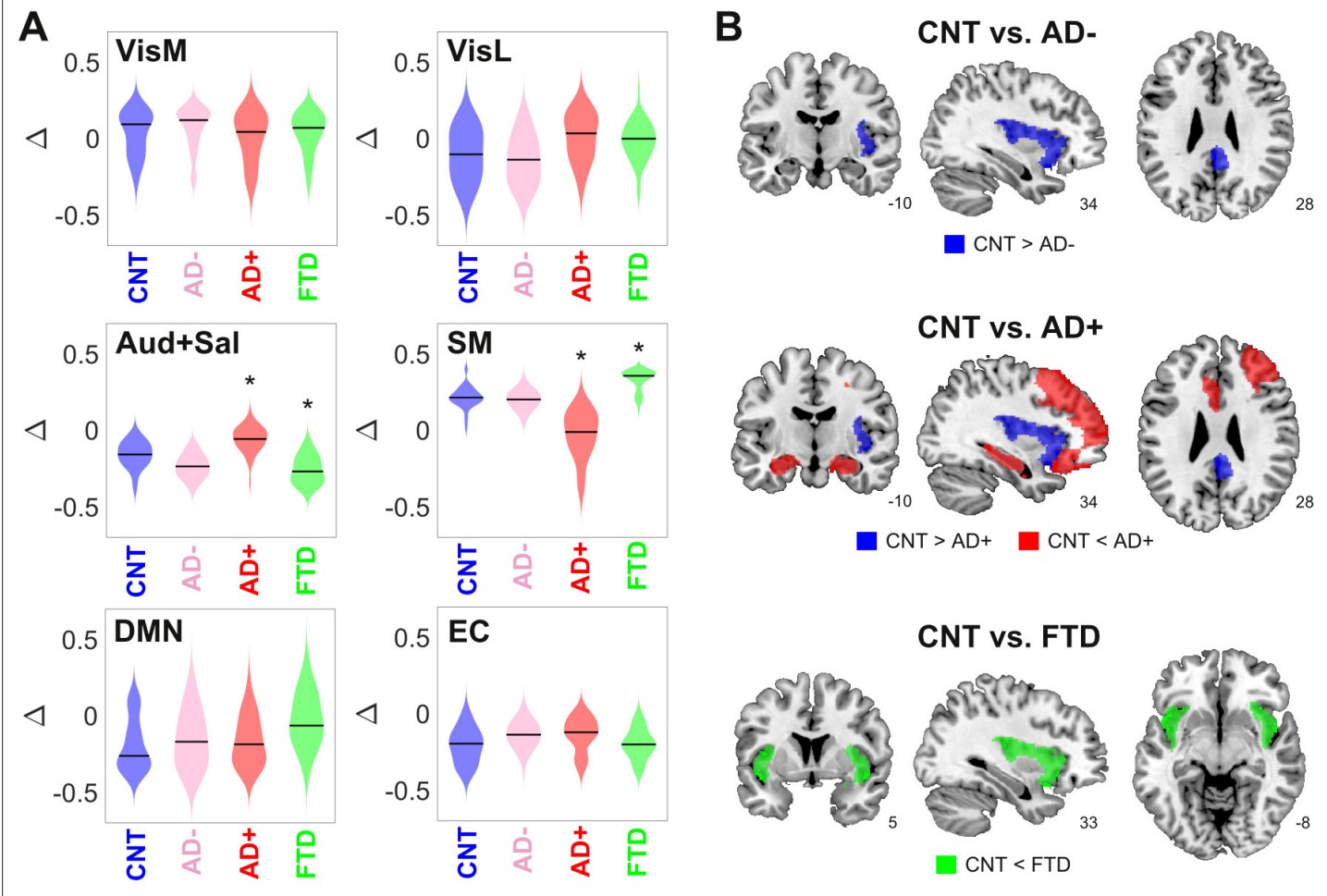

**Figure 3.** Changes in dynamical stability underlie differences in whole-brain functional connectivity (FC) between controls and patients diagnosed with neurodegenerative diseases. (**A**) Distribution of parameters Δ (which encode the contribution of each resting state networks (RSN) towards the local bifurcation parameter) across 300 runs of parameter optimization for CNT, Alzheimer's disease (AD-), AD+, and behavioral variant frontotemporal dementia (bvFTD) (* indicates $|d| > 0.8$ between distributions). (**B**) Brain regions with values $|d| > 0.8$ between the local distribution of bifurcation parameters of CNT vs. AD-, AD+, and bvFTD.

For this purpose, we used the median white-matter hyperintensity (WMHI) scores, a marker of small vessel cerebrovascular damage associated with AD disease severity, scored using fluid-attenuated inversion recovery (FLAIR) images (*Lee et al., 2016*). Vascular compromise has been recently identified as a critical selective vulnerability to AD in vascular subpopulations, transcriptomic perturbations, and expression of genes identified in AD genome-wide association studies (*Yang et al., 2022*). We chose to use an anatomical marker of disease severity instead of a cognitive/behavioral one since it is more closely related to the empirical neuroimaging data used to train the model. We compared the optimal model parameters (Δ and bifurcation parameters) between CNT, AD-, AD+, and bvFTD across 300 independent runs of the optimization procedure (*Figure 3A*). Comparing the resulting distributions in terms of the mean and standard deviation, we observed increased/decreased Δ values for the auditory-salience RSN (Aud +Sal) for AD/bvFTD, respectively, and the converse result for the sensorimotor network.

To facilitate the interpretation of these findings, we combined the local Δ values for each AAL, yielding the local bifurcation parameters; next, we visualized brain regions with increased or decreased values relative to control (defined as those regions with $|d| > 0.8$ between both distributions). The corresponding results are shown in *Figure 3B*. For AD patients with low WMHI scores (AD-), we only observed a shift towards dynamical instability (i.e., closer to the bifurcation) in the posterior cingulate cortex and the right insula. These results were maintained for the AD+ group, but we also observed

a shift away from the bifurcation towards stable fixed-point dynamics in the bilateral hippocampus and regions of the frontal cortex. Finally, the bvFTD group was characterized by a shift away from the bifurcation but in the opposite sense, that is, towards the synchronized regime, located in the bilateral insular cortex.

Overall, these results show that the changes in FC observed in AD and bvFTD can be related to region-specific shifts away from the complex and flexible dynamics that manifest in the proximity of the bifurcation point, with shifts towards stable noisy dynamics in the bilateral hippocampus for AD (but only for the subgroup with high WMHI scores) and shift towards stable oscillations in the bilateral insula for bvFTD. Note that these results concern the bifurcation parameters, which are obtained as a combination of the Δ values optimized using the genetic algorithm (*Figure 3A*).

### Latent space encoding

After investigating the differences between groups in the six-dimensional parameter space spanned by the Δ values, we obtained two-dimensional representations of these states by encoding the simulated FC using a VAE. *Figure 4A* presents a comparison between the original (above diagonal) and reconstructed (below diagonal) FC matrices; clearly, the reconstructed matrices closely resemble the originals. SSIM values between original and reconstructed FC matrices were 0.86 (CNT), 0.71 (AD+), 0.82 (AD-), and 0.76 (bvFTD). Panel B illustrates the structure of the FC matrix decoded at each point of the two-dimensional latent space spanned by the two hidden units of the VAE. Panel C shows the encoding of 300 independent runs of the model fitted to CNT, AD-, AD+, and bvFTD, with the larger colored circles indicating the average positioning of all runs in latent space. The two different neurodegenerative diseases, AD and bvFTD, were encoded in different directions of the latent space, indicating their qualitatively different effects on whole-brain connectivity and dynamics. Moreover, we observed that AD- and AD+ are within the same line in latent space, with AD- being closer to CNT than AD+, as expected in terms of the WHMI scores (see *Figure 4D*). These results highlight that the organization of the latent space is suggestive of different pathophysiological mechanisms and that it is sensitive to disease severity, which can be inferred from the relative placement of the encoded FC matrices.

### Perturbational landscapes of neurodegenerative diseases

The latent space encoding facilitates the visualization of complex manipulations applied to the model. We leveraged this method to investigate how each disease group responded to three different perturbations: periodic forcing at the peak BOLD frequency of each node (Wave stimulation), shift of the bifurcation parameter towards the synchronized regime (Sync perturbation), and shift towards the noisy regime (Noise perturbation). These perturbations can be applied with different intensities and at different pairs of homotopic brain regions, resulting in a series of FC matrices that depend on these two variables. After encoding, these FC matrices appear as a set of continuous trajectories in latent space that start from the state under perturbation (perturbational landscape), as can be seen in *Figure 5*. Interestingly, all perturbation protocols presented an asymptotic behavior where further increases of the parameter failed to keep displacing the encoded state. Here, the trajectories in dark gray tones are the closest to CNT, with lighter colors indicating less proximity. Both the Wave and Sync stimulations resulted in a similar perturbational landscape, as expected from the similar effects of these perturbations on the nodal dynamics. As shown in the right column of *Figure 5*, while AD- could be displaced to reach CNT, this became progressively more difficult for AD+ and bvFTD (for all pairs of distributions shown in this panel, $|d| > 0.8$). For bvFTD, in particular, very few trajectories approached the region corresponding to CNT.

### Brain regions that increased similarity to controls when stimulated

Finally, we ranked regions according to the proximity of their corresponding trajectories to CNT. It is important to note that the 2D location in the embedding and its proximity to CNT provides more information than one-dimensional metrics such as the GOF between the perturbed FCs and the CNT FC, including the trajectory of the perturbation (see *Figure 5—figure supplement 1* in the supplementary material). *Figure 6* presents this information, showing the top decile of regions for each disease subgroup and stimulation type. We found that Wave and Sync stimulation applied to regions in the visual and temporal lobes (mainly in the hippocampus) displaced both AD- and AD+

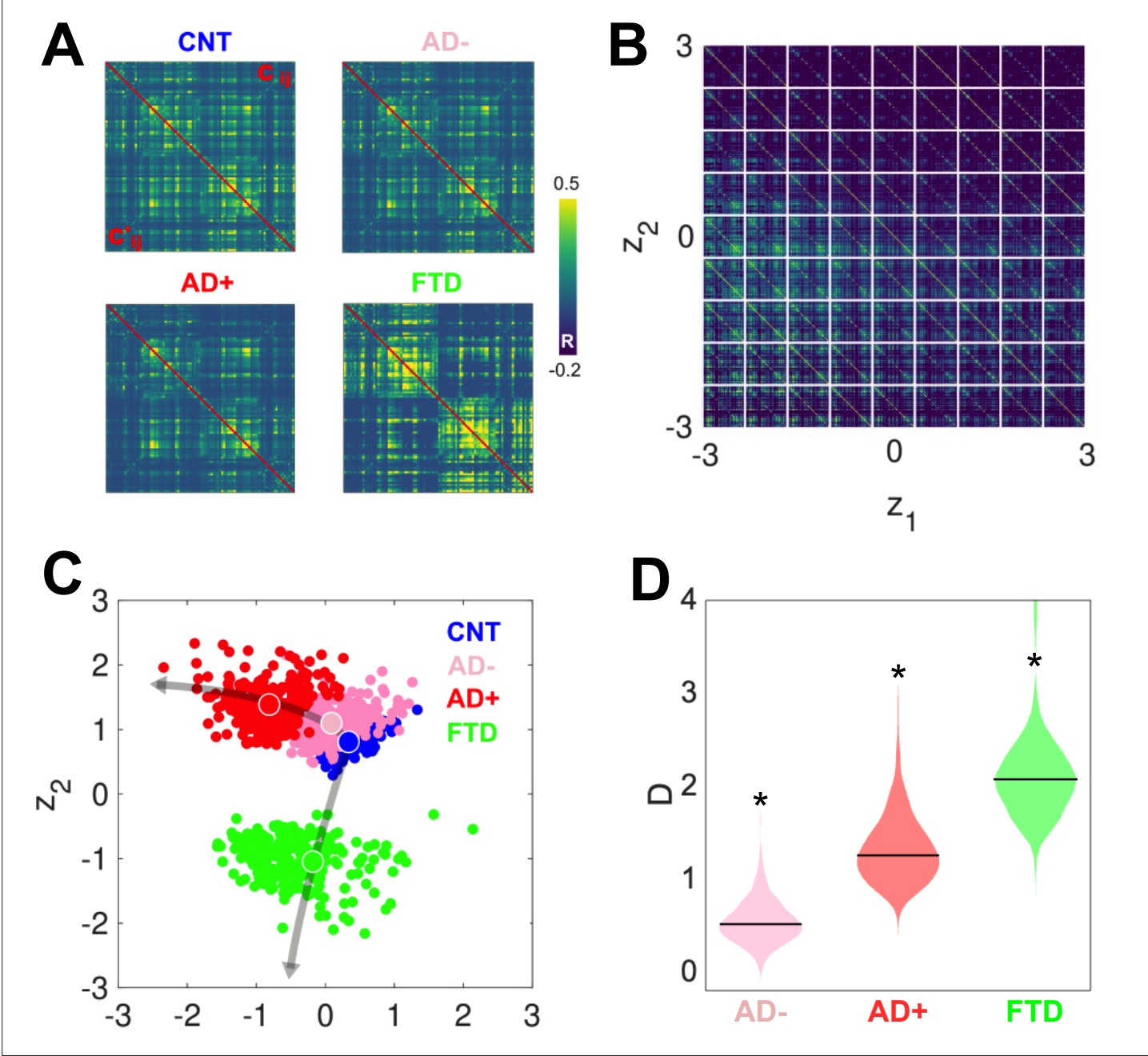

**Figure 4.** Latent space encoding of whole-brain functional connectivity (FC) reflects the different neurodegenerative diseases and the severity of Alzheimer's disease (AD) in terms of white-matter hyperintensity (WHMI) scores. (**A**) Original (above diagonal) and reconstructed (below diagonal) FC matrices. (**B**) Illustration of the FC matrix encoded at each point of the latent space by the variational autoencoder (VAE). (**C**) Latent space encoding of 300 independent runs of the model fitted to empirical CNT, AD-, AD+, and behavioral variant frontotemporal dementia (bvFTD) data. The larger circles represent the average of the positioning of all encoded points for each group. (**D**) Distribution of distances (D) to CNT for each condition (*indicates $|d|$ >0.8 between distributions).

towards CNT. In the case of bvFTD, frontal regions involved with social cognition were highlighted. For the Noise protocol, regions of the sensorimotor cortex were the most important to displace the brain state from AD to CNT, while frontal regions were again the most relevant for the bvFTD group. Finally, for AD the temporo-posterior involvement was more systematic across different types of stimulation compared to bvFTD.

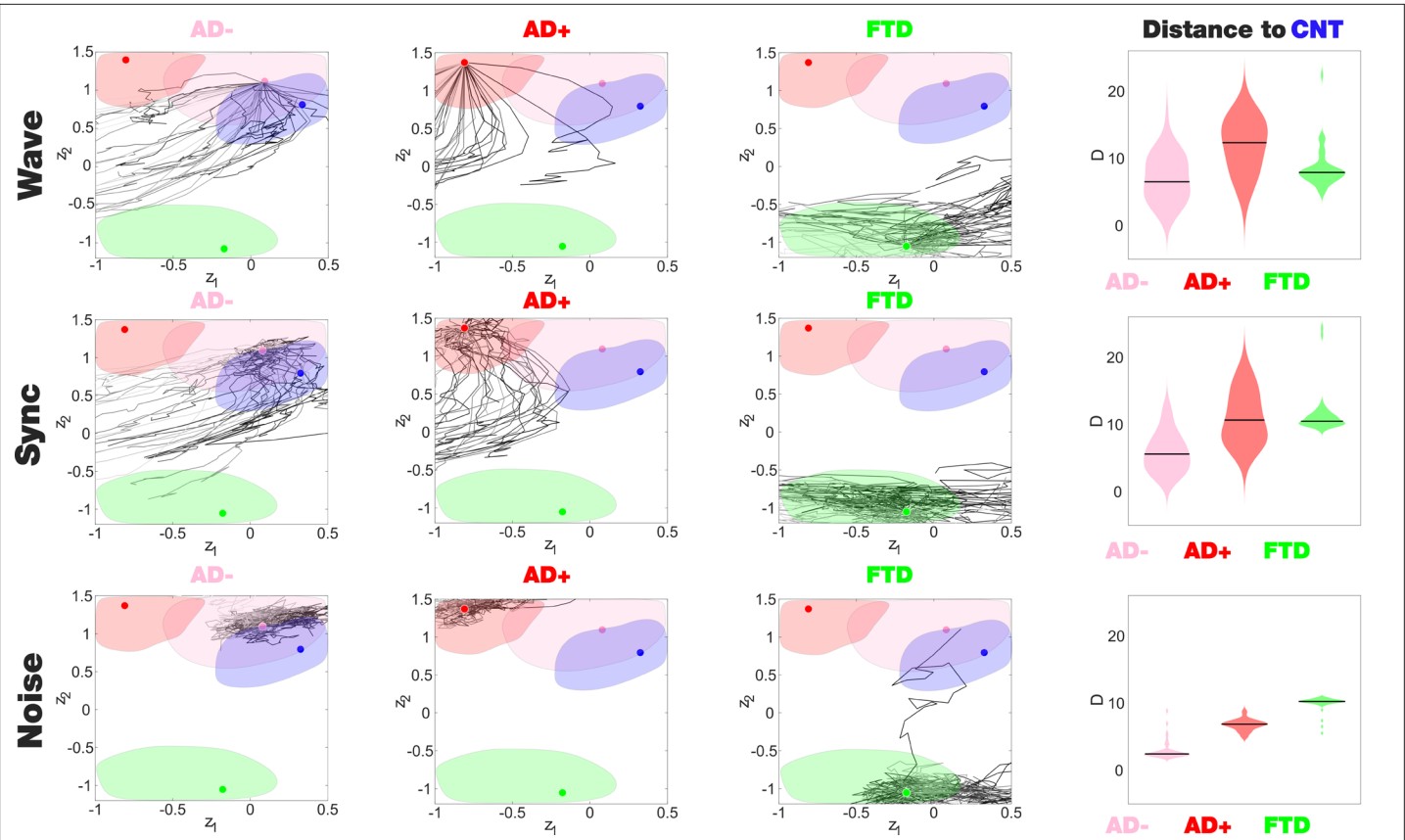

**Figure 5.** Perturbational landscapes of neurodegenerative diseases. Each row corresponds to a different simulated stimulation protocol (Wave, Sync, and Noise), while each column corresponds to a different group (Alzheimer's disease [AD-], AD+, and behavioral variant frontotemporal dementia [bvFTD]). Trajectories represent the encoded sequence of functional connectivity (FC) matrices obtained as a result of parametrically increasing the perturbation. Individual trajectories represent the outcome of the stimulation applied at different pairs of homotopic Automated Anatomical Labeling (AAL) regions, with darker tones indicating higher proximity to the target state, CNT. The rightmost column presents the minimum distances of the trajectories to CNT, with AD- being the closest in this sense.

The online version of this article includes the following figure supplement(s) for figure 5:

**Figure supplement 1.** Mapping 1 – GOF in the latent space comparing with distance to CNT.

## Discussion

We used a phenomenological model of fMRI dynamics informed by disease-specific priors to investigate the mechanisms underlying whole-brain FC changes in AD and bvFTD, the most common neurodegenerative diseases leading to dementia (*Bang et al., 2015*; *Scheltens et al., 2021*). The use of VAE for dimensionality reduction allowed us to explore the global effects of simulated stimulation in the model, yielding the perturbational landscape of AD and bvFTD whole-brain dynamics, from which we identified key regions to induce transitions towards healthy brain states. In the following, we discuss our findings in light of the multiscale pathophysiology of neurodegenerative diseases and the different interventions that have been proposed for its treatment.

Contrary to previous implementations of the Hopf normal mode as a whole-brain activity model, we allowed regional variations constrained by different spatial heterogeneity maps. We obtained the best fits using priors based on RSN and disease-specific atrophy maps, with lower GoF when using maps obtained from a different neurodegenerative disease (PD), prompting the need to discuss the relationship between atrophy and local dynamics. AD is linked to altered cellular energy metabolism (*Gu et al., 2012*), excitation/inhibition ratio (*Lopatina et al., 2019*; *Maestú et al., 2021*) and neurotrophic factor release (*Murer et al., 2001*), impairing neural microcircuit function (*Palop and Mucke, 2016*). Using TMS, it was shown that imbalances between inhibition and excitation correlate with negative and positive symptoms in bvFTD patients (*Benussi et al., 2020b*), and distinguish between

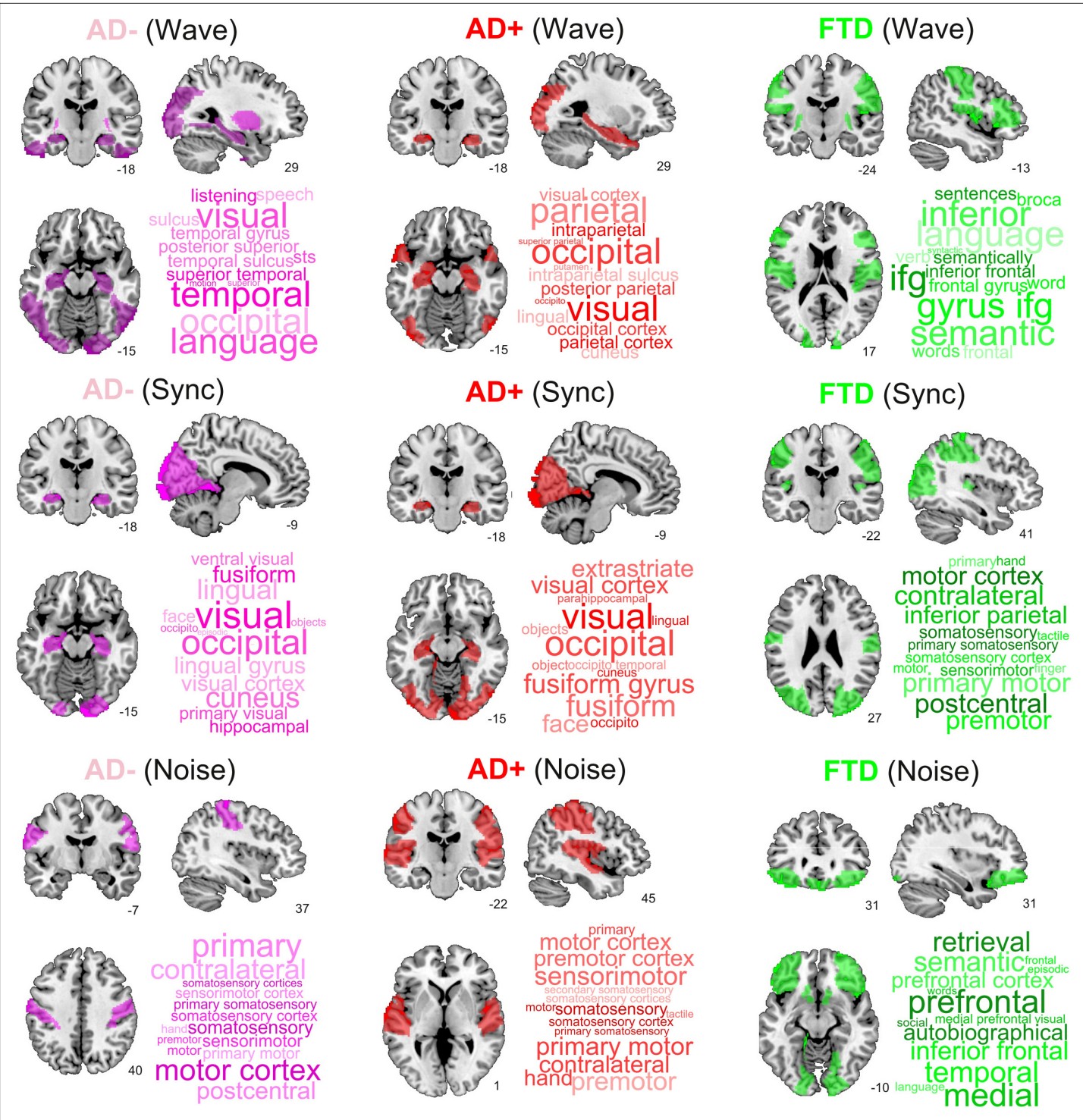

**Figure 6.** Top brain regions ranked according to the proximity of their corresponding trajectories to CNT. Columns correspond to disease subgroups (Alzheimer's disease [AD-], AD+, and behavioral variant frontotemporal dementia [bvFTD]), while rows correspond to the three explored simulated stimulation protocols (Wave, Sync, and Noise).

bvFTD and AD patients (*Benussi et al., 2017*). Considering the relationship between the Hopf bifurcation parameter and the local excitation/inhibition ratio (which can be inferred by studying the bifurcation diagram of more biophysically realistic models, such as the Wilson–Cowan model) (*Cowan et al., 2016*), it is likely that this feature of cortical dynamics is being captured during local parameter

optimization. The methodological framework developed here could be applied to investigate the dynamical consequences of other spatial heterogeneities, which in the case of AD and bvFTD include glucose metabolism, tau and amyloid PET imaging, and multimodal allostatic markers (*Engler et al., 2008*; *Foster et al., 2007*; *Jack et al., 2018*; *Mosconi et al., 2008*; *Nordberg, 2004*; *Migeot et al., 2022*).

The regions with the most prominent alterations in the local bifurcation parameters are the most affected by AD and bvFTD pathology. Temporo-posterior atrophy, including the hippocampus in the medial temporal lobe (MTL), is indicative of AD and predicts disease onset and progression (*Pini et al., 2016*). Moreover, these regions also show altered FC profiles (*Liu et al., 2008*) and hemodynamic responses in spatial navigation (*Vlček and Laczó, 2014*) and memory-related tasks (*Golby et al., 2005*), both functions compromised in AD. The expansion of temporo-posterior compromise to more frontal regions in AD+ is also consistent with multimodal models of disease progression (*Raj et al., 2015*). The shift towards stable fixed-points hippocampal dynamics is consistent with reports of functional uncoupling of the MTL in AD (*Berron et al., 2020*), as well as with increased cortical inhibition (*Maestú et al., 2021*). On the other hand, bvFTD presents disruptions in a fronto-insular-temporal network implied in social cognition, among other functions (*Beeldman et al., 2018*; *Ibáñez, 2018*; *Legaz et al., 2022*; *Salamone et al., 2021*; *Santamaría-García et al., 2017*). In particular, the breakdown of the salience network includes degeneration of the anterior insula, impacting emotional, social, and interoceptive deficits in bvFTD (*Garcia-Cordero et al., 2021*; *Ibañez and Manes, 2012*; *Migeot et al., 2022*). Consistent with our results (i.e., shift away from the bifurcation towards synchronous dynamics in the bilateral insula), these abnormalities manifest in the fMRI signal as increases in the power of low-frequency oscillations (*Day et al., 2013*). Interestingly, using the RSN prior resulted in GoF values comparable to those obtained using AD atrophy maps, a result compatible with the observation that disrupted whole-brain dynamics and FC in AD patients follows the general outline of the major RSN (*Brier et al., 2012*).

While previous modeling studies addressed the effect of localized perturbations (*Deco et al., 2019*; *Ipina et al., 2020*; *Leonardo et al., 2017*), the high dimensionality of whole-brain FC has hampered their global analysis and visualization. For this purpose, we adopted a framework based on VAE, a deep learning method for nonlinear dimensionality reduction, capable of representing the simulated FC matrices in a bidimensional latent space with low reconstruction error (*Perl et al., 2020*). Within this latent space, we distinguished controls, AD and bvFTD patients, with different directions encoding the severity of each disease. Despite the distinctive clinical patterns that characterize these two diseases, their overlap in terms of cognition and behavior can complicate their differential diagnosis (*Musa et al., 2020*; *Parra et al., 2018*). Our model was capable of learning FC patterns that mapped into clearly separated latent space regions, suggesting that future modeling efforts at the single subject level could significantly contribute to the problem of classification between AD and bvFTD, assisting the development of mechanistic biomarkers (*Deco and Kringelbach, 2014*). Moreover, the latent space encoding also captured the severity of AD in terms of WMHI (note that the same analysis could not be conducted for the bvFTD dataset due to insufficient number of participants). Also, while WMHI are highly prevalent and associated with disease severity, such association is weak in the case of bvFTD patients (*Hu et al., 2021*). As documented in the literature, the severity of small-vessel cerebrovascular disease (visualized using T2-weighted MRI) is increased in AD patients, serves as a predictor of AD future development and rate of cognitive decline, and correlates with atrophy progression (*Hu et al., 2021*; *Kim et al., 2020*; *Lee et al., 2016*; *Liu et al., 2018*). Moreover, our results support recent critical evidence unveiling a major vascular contribution to AD (*Yang et al., 2022*). The low (AD-) and high (AD+) severity groups were encoded within the same direction, but at different distances from the control group, suggesting that disease progression could be described in terms of a one-dimensional manifold.

We explored three different forms of simulated stimulation, finding one trajectory in latent space per pair of homotopic AAL regions (where stimulation was delivered), parameterized by $\Delta a$ (for Noise and Sync stimulation), and $F_0$ (for Wave stimulation). As expected, the Wave stimulation delivered at the dominant frequency of each node displaced the dynamics towards the oscillatory regime, resulting in effects very similar to those obtained with the Sync stimulation. For these two types of stimulation, key nodes to transition from AD towards the healthy state included the hippocampus as well as temporo-posterior regions. In the case of bvFTD, these regions comprised frontal areas

involved with social cognition, which is known to be compromised by this pathology (**Bang et al., 2015**; **Garcia-Cordero et al., 2021**; **Ibañez and Manes, 2012**; **Migeot et al., 2022**). Interestingly, the brain regions important to revert abnormal brain FC did not match with those presenting different bifurcation parameters (see **Figure 3**), implying that the simulated perturbation is not limited to the normalization of these differences, but also causes complex downstream effects displacing the dynamics in the desired direction.

Our study presents some limitations to be acknowledged, partly stemming from the model and the sources of empirical data necessary for its construction. Phenomenological models are not directly interpretable in terms of biophysical variables, and they only allow exploring mechanisms associated with general aspects of the dynamics. With respect to model fitting, as in previous studies (**Ipina et al., 2020**; **Sanz Perl et al., 2021**; **Perl et al., 2020**), we observed low inter-hemispheric simulated FC, which is related to the known underestimation of homotopic fiber tracts by DTI (**Reveley et al., 2015**). These analyses should be revisited when more accurate whole-brain connectivity estimates become available, expecting a better fit to the empirical data that could strengthen the main conclusions of our work. We conducted group-level analyses, which constitute a valuable proof of principle for applying whole-brain modeling to neurodegenerative diseases, but fall short of providing information at the single subject level, which would be required to correlate the results of the model with behavioral outcomes, especially given results showing that the therapeutic effect of transcranial stimulation depends on individual characteristics of brain anatomy (**Unal et al., 2020**; **Tsapkini et al., 2018**; **de Aguiar et al., 2020**; **Pytel et al., 2021**). For this purpose, models tailored to data from individual patients will be required (**Menardi et al., 2022**). Another limitation of this work is related to the diagnosis of the patients, which was based on current clinical criteria. Mainstream dementia diagnosis may also include biomarkers such as deposition of amyloid-β and tau proteins. These are quantified through positron emission tomography (PET) or plasma, especially in AD (**Agrawal and Biswas, 2015**). However, it is currently common for dementia research to use the standard clinical criteria as reported in this paper (**Birba et al., 2022**; **Herzog et al., 2022**; **Legaz et al., 2022**; **Maito et al., 2023**; **Parra et al., 2023**). Moreover, PET/plasma biomarkers present caveats for global settings. PET is not widely available, that is, the percentage of PET access for patients in global settings is less than 1% (**Parra et al., 2023**; **Parra et al., 2018**) nor cost-effective (**Parra et al., 2023**). It does not provide a conclusive diagnosis (**Brown et al., 2014**) nor discriminates well between FTD variants (**Ntymenou et al., 2021**). Fluid biomarkers (i.e., plasma) are very promising, but are not yet massively accessible. More importantly, plasma biomarkers currently lack systematic validation in diverse and nonstereotypical populations (**Parra et al., 2023**) like the current sample. However, future works should (a) combine clinical and biomarkers criteria to model whole brain dynamics and (b) use metabolic maps of tau and amyloid maps as priors to improve the fit of the whole brain modeling. Finally, despite our choice of the AAL parcellation being based on its widespread adoption in dementia research (**Agosta et al., 2013**; **Lord et al., 2016**; **Reyes et al., 2018**; **Sedeño et al., 2017**; **Whitwell et al., 2011**), future research may require a systematic exploration of other brain parcellations applied to the dementia population. In that direction, to compare the differences across parcellations, a recent study by Gonzalez et al. used the AAL and HCP atlas (**Glasser et al., 2016**) parcellations on a dementia subsample (graph connectivity and graph multifeature in both modalities) and did not find significant differences (**Gonzalez-Gomez et al., 2023**).

In conclusion, we developed a computational model capable of capturing the most salient differences in whole-brain dynamics between controls and patients. In doing so, the model informed potential underlying dynamical mechanisms that could be traceable to biophysical observables, such as the local excitation/inhibition ratio. We also introduced a novel methodological framework to visualize complex global manipulations of brain activity, leveraging it to investigate the relationship between bvFTD and AD of different severity, as well as their perturbational landscape. Whole-brain models of neurodegeneration may bring together innovative combinations of multimodal brain biomarkers (atrophy, structural and functional connectivity, vascularity, metabolism) while assessing the critical challenges of dimensionality, heterogeneity, and differential diagnosis. While the implemented model facilitated interpretation in dynamical terms, future efforts should expand our work towards more biophysically realistic models, allowing the assessment of pharmacological modulation, among other simulated interventions to enhance cognitive function and slow down its decline in patients suffering from neurodegeneration.

## Materials and methods

### Experimental design

This is a modeling study that includes multiple sources of empirical data (DTI, fMRI, structural MRI, and FLAIR imaging), with the objective of investigating the dynamical mechanisms underlying structural and functional changes in patients diagnosed with neurodegenerative diseases (AD and bvFTD), and of searching for optimal external stimulation protocols capable of restoring dynamics indicative of healthy participants.

### Participants

The study comprised 94 participants: 39 patients diagnosed with AD (26 females, 76.6 ± 7 y [mean ± SD]), 18 patients diagnosed with bvFTD (5 females, 66.7 ± 10.8 y), and 57 healthy controls (38 females, 69.8 ± 7.9 y), no significant differences in the age of participants were observed between groups (Kruskal–Wallis test, p>0.05). Additionally, a group of 43 PD patients (18 females, 68.7 ± 8.2 y) was used to obtain the atrophy maps for this condition. Patients were diagnosed by expert neurologists following current criteria for probable behavioral variant bvFTD (*Rascovsky et al., 2011*), and NINCDS-ADRDA clinical criteria for AD (*McKhann et al., 1984*; *McKhann et al., 2011*). Recruitment and diagnosis were conducted in clinical centers by a multidisciplinary team as part of an ongoing multicentric protocol (*Donnelly-Kehoe et al., 2019*; *Salamone et al., 2021*; *Salamone et al., 2021*). Diagnoses were supported by extensive examinations (*Baez et al., 2014*; *Melloni et al., 2016*; *Piguet et al., 2011*; *Santamaría-García et al., 2017*), in line with the Multi-Partner Consortium to Expand Dementia Research in Latin America (ReDLat) standardized protocol (*Ibanez et al., 2021a*; *Ibanez et al., 2021b*; *Maito et al., 2023*). Fifty-four participants fulfilled the NINCDS-ADRDA criteria for typical AD (*Dubois et al., 2007*) presented memory deficits and showed atrophy in middle-temporal, hippocampal, and posterior regions, among others (*Figure 2—figure supplement 1* in the supplementary material). Thirty-one participants met the revised criteria for probable bvFTD (*Piguet et al., 2011*), presented behavioral and social deficits according to caregivers, and showed fronto-temporo-insular atrophy (see *Figure 2—figure supplement 1* and detailed explanation on the atrophy maps computation in the supplementary material). No participants reported a history of other neurological disorders, psychiatric conditions, primary language deficits, or substance abuse. All participants provided written informed consent pursuant to the Declaration of Helsinki. The study was approved by the Ethics Committees of the involved institutions (Comite de Ética Científico Servicio de Salud Metropolitano Oriente, IRB00007701; Instituto Alberto C. Taquini de Investigaciones en Medicina Traslacional, IRB00013030; Universidad Adolfo Ibañez, IRB00012394).

### MRI data acquisition

We acquired three-dimensional volumetric and 10-min-long resting-state MRI sequences. Participants were instructed not to think about anything in particular while remaining still, awake, and with eyes closed. Recordings were conducted in two independent centers, using the parameters described below.

#### Center 1 (Argentina)

Using a 3 T Phillips scanner with a standard head coil, we acquired whole-brain T1-rapid anatomical 3D gradient echo volumes, parallel to the plane connecting the anterior and posterior commissures, with the following parameters: repetition time (TR) = 8300 ms; echo time (TE) = 3800 ms; flip angle = 8°; 160 slices, matrix dimension = 224 × 224 × 160; voxel size = 1 mm × 1 mm × 1 mm. Also, functional spin echo volumes, parallel to the anterior-posterior commissures, covering the whole brain, were sequentially and ascendingly acquired with the following parameters: TR = 2640 ms; TE = 30 ms; flip angle = 90°; 49 slices, matrix dimension = 80 × 80 × 49; voxel size in plane = 3 mm × 3 mm ×3 mm; slice thickness = 3 mm; sequence duration = 10 min; number of volumes = 220. A total of 18 AD patients, 13 bvFTD patients, and 23 controls were scanned in this center.

#### Center 2 (Chile)

Using a 3 T Siemens Skyra scanner with a standard head coil, we acquired whole-brain T1-rapid gradient echo volumes, parallel to the plane connecting the anterior and posterior commissures,

with the following parameters: repetition time (TR) = 2400 ms; echo time (TE) = 2000 ms; flip angle = 8°; 192 slices, matrix dimension = 256 × 256 × 192; voxel size = 1 mm × 1 mm × 1 mm. Finally, functional EP2D-BOLD pulse sequences, parallel to the anterior-posterior commissures, covering the whole brain, were acquired sequentially intercalating pair-ascending first with the following parameters fMRI parameters: TR = 2660 ms; TE = 30 ms; flip angle = 90°; 46 slices, matrix dimension = 76 × 76 × 46; voxel size in plane = 3 mm × 3 mm × 3 mm; slice thickness = 3 mm; sequence duration = 10.5 min; number of volumes = 240.

## Anatomical connectivity

Anatomical connectivity was obtained applying diffusion tensor imaging (DTI) to diffusion weighted imaging (DWI) recordings from 16 healthy right-handed participants (11 men and 5 women, mean age: 24.75 ± 2.54 y) recruited at Aarhus University, Denmark. For each participant, a 90 × 90 matrix was obtained, representing the connectivity between pairs of AAL regions. Data preprocessing was performed using FSL diffusion toolbox (Fdt) with default parameters. The probtrackx tool in Fdt provided automatic estimation of crossing fibers within voxels, which has been shown to significantly improve the tracking sensitivity of non-dominant fiber populations in the human brain. The connectivity probability from a seed voxel $i$ to another voxel $j$ was computed as the proportion of fibers passing through voxel $i$ that reached voxel $j$ (sampling of 5000 streamlines per voxel). All the voxels in each AAL region were seeded (both gray and white matter voxels were considered). The connectivity probability $P_{ij}$ from region $i$ to region $j$ was calculated as the number of sampled fibers in region $i$ that connected the two regions, divided by 5000 × n, where n is the number of voxels in region $i$. The resulting connectivity matrices were computed as the average across voxels within each region in the AAL, thresholded at 0.1% (i.e., a minimum of five streamlines) and normalized by the number of voxels in each ROI. Finally, the data were averaged across participants to yield the $K_{ij}$.

## fMRI data preprocessing

We discarded the first five volumes of each fMRI resting state recording to ensure a steady state. Images were preprocessed using the Data Processing Assistant for Resting-State fMRI (DPARSF V2.3) (*Chao-Gan and Yu-Feng, 2010*), an open-access toolbox that generates and implements an automatic pipeline for fMRI analysis within SPM12 and the Resting-State fMRI Data Analysis Toolkit (REST V.1.7) (*Song et al., 2011*). Preprocessing steps included slice-timing correction and realignment to the first scan of the session to correct head movement. Using least squares, we regressed out six motion parameters, as well as cerebrospinal fluid and white matter signals to attenuate the potential effects of residual movement and physiological noise. For this purpose, motion parameters were estimated during the realignment step, and cerebrospinal fluid and white matter masks were obtained from the tissue segmentation of each subject's T1 scan in native space. None of the participants presented head movements larger than 3 mm and/or rotations higher than 3° and no differences in head motion among groups were found. As a final step, images were normalized to common MNI space and smoothed using an 8 mm full-width-at-half-maximum isotropic Gaussian kernel.

## Atrophy maps

Images were preprocessed using the DARTEL Toolbox. After smoothing with a 10 mm full-width half-maximum kernel, images were normalized to the MNI space and analyzed through general linear models for 2nd level analyses on SPM-12 software. To analyze the images of each center together and avoid biases due to different scanners in our results, the normalized and smoothed outputs were transformed to W-score images adjusted for age, disease, total intracranial volume and scanner type. W-scores, similar to Z-scores (mean = 0, SD = 1), represent the degree to which the observed gray matter volume in each voxel is higher or lower (positive or negative W-score) than expected relative to the healthy control sample of each acquisition center.

## White matter hyperintensity

From raw FLAIR images, white matter lesions were segmented using the lesion prediction algorithm (LPA) as implemented in the Lesion Segmentation Toolbox version 2.0.15 (https://www.statistical-modelling.de/lst.html) for SPM (SPM12, Matlab v.2020a; MathWorks, Natick, MA), based on the calculation of a lesion probability score for each voxel. Lesion probability maps were smoothed using a

Gaussian kernel with FWHM at 1 mm for voxels with a lesion probability >0.1. Voxels with no direct neighbors were deleted from the lesion maps. Lesion size maps were then acquired from the probability maps considering voxels with a probability >0.5 and lesion sizes with a threshold >0.015. Lesion maps were visually inspected per subject to check for possible artifacts and discarded when artifacts were found (in choroid plexus and in basal cisterns). Total WMHI volume in cubic centimeters was defined as the voxel size multiplied by the total number of voxels labeled as lesions in the cerebrum. The total WMHI volume was normalized by the total intracranial volume in each subject.

## Group averaged FC matrices

fMRI signals were detrended, demeaned, and band-pass filtering in the 0.04–0.07 Hz range. This frequency range was chosen because it was shown to contain more reliable and functionally relevant information compared to other bands, and to be less affected by noise (**Cordes et al., 2001**). Subsequently, the filtered time series were transformed to z-scores. Fixed-effect analysis was used to obtain group-level FC matrices, meaning that the Fisher's R-to-z transform ($z = \mathrm{atanh}\,(R)$) was applied to the correlation values before averaging over participants within each group.

## Model equations

The whole-brain model consisted of nonlinear oscillators coupled by the structural connectivity matrix, $K_{ij}$. Each oscillator was modeled using the normal form of a Hopf bifurcation, which represented the dynamics at one of the 90 brain regions in the AAL template (**Tzourio-Mazoyer et al., 2002**). We adopted the neurobiological assumption that dynamics of macroscopic neural masses range from fully synchronous to a stable asynchronous state governed by random fluctuations. Building upon previous work, we also assumed that fMRI can capture the dynamics from both regimes with sufficient fidelity to be modeled by the equations (**Deco et al., 2019**; **Deco et al., 2017**; **Demirtaş et al., 2017**; **Sanz Perl et al., 2021**). Passing through the Hopf bifurcation changes the qualitative nature of the solutions from a stable fixed point in phase space towards a limit cycle, allowing the model to present self-sustained oscillations. Without coupling, the local dynamics of brain region j was modeled by the complex equation:

$$\frac{dz_j}{dt} = \left[a + i\omega_j\right] z_j - z_j |z_j|^2 \tag{1}$$

Here, $z_j$ is a complex-valued variable ($z_j = x_j + iy_j$), and $\omega_j$ is the natural oscillation frequency of node j. These frequencies ranged from 0.04 to 0.07 Hz and were determined by the averaged peak frequency of the bandpass-filtered fMRI signals at each individual brain region. The parameter a represents the bifurcation parameter that controls the dynamical behavior of the system. For a < 0, the phase space presents a unique stable fixed point at $z_j = 0$ and the system asymptotically decays towards this point. For a > 0, the stable fixed point gives rise to a limit cycle with self-sustained oscillations of frequency $f_j = \omega_j/2\pi$ and amplitude proportional to $\sqrt{a}$. In the full model, nodes i and j are coupled by the structural connectivity matrix $C_{ij}$. To ensure oscillatory dynamics for a > 0, the SC matrix was scaled to a maximum of 0.2 (weak coupling condition) (**Deco et al., 2017**). In full form, the differential equations of the model are:

$$\frac{dx_j}{dt} = \frac{d\mathrm{Re}\,(z_j)}{dt} = \left[a - x_j^2 - y^2\right] x_j - \omega_j y_j + G \sum_{i=1}^{90} K_{ij}\,(x_i - x_j)$$
$$+ \beta \eta_j\,(t)$$

$$\frac{dy_j}{dt} = \frac{d\mathrm{Im}\,(z_j)}{dt} = \left[a - x_j^2 - y^2\right] y_j + \omega_j x_j + G \sum_{i=1}^{90} K_{ij}\,(y_i - y_j)$$
$$+ \beta \eta_j\,(t) \tag{2}$$

The parameter G represents a global coupling factor that scales the anatomical coupling equally for all node pairs, and $\beta_j$ represents the amplitude of the additive Gaussian noise in each node, which was fixed at 0.04. Note that when a is close to the bifurcation (a ~ 0) the additive Gaussian noise gives rise to complex dynamics as the system continuously switches between both sides of the bifurcation. For each choice of parameters, these equations were integrated using the Euler–Maruyama algorithm with a time step of 0.1 s.

## Spatial heterogeneity priors

Based on previous work (*Deco et al., 2018*; *Ipina et al., 2020*; *Sanz Perl et al., 2021*), we introduced additional parameters to account for regional variations in the dynamical regime of the nodes. Since introducing an independent bifurcation parameter for each node could lead to costly optimization and overfitting, we grouped AAL regions with the same contribution to the local parameters. This grouping procedure was conducted using different priors encoding the possible spatial heterogeneities that can be captured by the model.

Each prior consisted of a grouping matrix $M_{i,j}$, which had 1 in its $i,j$ entry if the region $i$ is in group $j$ (note that groups could be overlapping). Group $j$ contributed an independent coefficient to the bifurcation parameter of region $i$, given by $\Delta_{i,j}$, which was obtained by the linear combination:

$$a_i = \sum_{j=1}^{N} \Delta_{i,j} M_{i,j} \tag{3}$$

We explored five different priors: the RSN prior (grouping the nodes based on their membership to six RSN), the AD + FTD atrophy prior (divided the combined AD and bvFTD atrophy map into six ranges of values, resulting in equally sized groups), PD atrophy prior (same as for the AD + FTD atrophy prior), random assignment of nodes into six groups (Random prior), and assignment into six equally sized groups based on anatomical proximity (Equal prior).

## Parameter optimization

The objective is to fit the model to maximize the similarity between the simulated and empirical FC matrices. Following previous work, we computed the GoF using the structure similarity index (SSIM) (*Wang et al., 2004*), a metric that balances sensitivity to absolute and relative differences between the FC matrices. Thus, the SSIM can be considered a trade-off between the Euclidean and correlation distances. For further details on the computation of the SSIM, see *Sanz Perl et al., 2021*.

Next, the scaling parameter G was obtained by the exhaustive exploration of the model with spatially homogeneous bifurcation parameters. For this, the GoF between empirical and simulated FC was computed over a 100 × 100 grid in parameter space, with the bifurcation parameter a in the [–0.2, 0.2] interval and G in the [0,3] interval. Note that the simulated FC matrix was obtained using the procedure outlined in the 'Group averaged FC matrices' subsection (resampled to one sample per 2 s and bandpass filtered in the 0.04–0.07 Hz range). After averaging 50 independent runs, we found the absolute minimum of GoF in a = 0 and G = 0.5 for all three groups of subjects (controls, AD, and bvFTD). These results were used as initial conditions in the following model that incorporated regional variation in the bifurcation parameters, fixing G = 0.5 in further analyses.

Afterwards, the coefficients $\Delta_{i,j}$ remain to be optimized. For this purpose, we implemented a genetic algorithm inspired in biological evolution. This method consists of an algorithmic representation of natural selection, which lets the fittest individuals prevail in the next generation, thus spreading the genes responsible for their better fitness.

The algorithm starts with a generation of 10 sets of parameters ('individuals') chosen randomly with values close to zero, to then generate a population of outputs with their corresponding GoF. A score proportional to the GoF is assigned to each individual. Afterwards, a group of individuals is chosen based on this score ('parents'), and the operations of crossover, mutation, and elite selection are applied to them to create the next generation. These three operations can be described as follows: (1) elite selection occurs when an individual of a generation shows an extraordinarily high GoF in comparison to the other individuals, thus this solution is replicated without changes in the next generation; (2) the crossover operator consists of combining two selected parents to obtain a new individual that carries information from each parent to the next generation; and (3) the mutation operator changes one selected parent to induce a random alteration in an individual of the next generation. In our implementation, 20% of the new generation was created by elite selection, 60% by crossover of the parents, and 20% by mutation. A new population is thus generated ('offspring') that is used iteratively as the next generation until at least one of the following halting criteria is met: (1) 200 generations are reached (i.e., limit of iterations), (2) the best solution of the population remains constant for 50 generations, and (3) the average GoF across the last 50 generation is less than $10^{-6}$. Finally, the output of the genetic algorithm contains the simulated FC with the highest GoF, and the optimal coefficients $\Delta_{i,j}$.

## Simulated stimulation

We implemented three different simulated stimulation protocols to induce transitions between pathological and healthy states. As in previous work, all stimulations were applied to pairs of homotopic nodes (*Deco et al., 2019*; *Sanz Perl et al., 2021*). The Wave stimulation corresponds to an additive periodic forcing term incorporated to the equation of the nodes, given by $F_0 \cos(\omega_0 t)$, where $F_0$ is the forcing amplitude and $\omega_0$ the dominant frequency of the nodes, obtained as explained in the 'Model equations' subsection. The Sync stimulation corresponds a change in the local bifurcation parameters given by $\Delta a > 0$, that is, towards increased synchronization. The Noise stimulation corresponds to a change in the local bifurcation parameters given by $\Delta a < 0$, that is, towards fixed-point noisy dynamics.

For each pair of homotopic regions, the parameter representing the strength of the perturbation (either $F_0$ or $\Delta a$) was increased from 0 to 2 in steps of 0.1 (averaging 100 independent simulations for each node pair). For the Noise stimulation, the parameter $\Delta a$ was decreased from 0 to –2. The matrices obtained for each value of the stimulation strength parameters and choice of stimulation regions were encoded in latent space using the VAE, as described in the next subsection.

## Encoding and decoding with VAE

We implemented a VAE to encode the FC matrices $C_{ij}$ in a low-dimensional representation. VAE map inputs to probability distributions in latent space, which can be regularized during the training process to produce meaningful outputs after the decoding step, allowing to decode latent space coordinates. The architecture of the implemented VAE (shown in *Figure 1*) consists of three parts: the encoder network, the middle variational layer, and the decoder network. The encoder consists of a deep neural network with rectified linear units (ReLu) as activation functions and two dense layers. This part of the network bottlenecks into the two-dimensional variational layer, with units $z_1$ and $z_2$ spanning the latent space. The encoder network applies a nonlinear transformation to map the $C_{ij}$ into Gaussian probability distributions in latent space, and the decoder network mirrors the encoder architecture to produce reconstructed matrices $C_{ij}^*$ from samples of these distributions.

Network trained consists of error backpropagation via gradient descent to minimize a loss function composed of two terms: a standard reconstruction error term (computed from the units in the output layer of the decoder), and a regularization term computed as the Kullback–Leibler divergence between the distribution in latent space and a standard Gaussian distribution. This last term ensures continuity and completeness in the latent space, that is, that similar values are decoded into similar outputs, and that those outputs represent meaningful combinations of the encoded inputs.

We generated 5000 correlation matrices $C_{ij}$ corresponding to controls, AD and bvFTD, using the model optimized as described in the 'Parameter optimization' subsection. We then created 80%/20% random splits into training and test sets using the training set to optimize the VAE parameters. The training procedure consisted of batches with 128 samples and 50 training epochs using an Adam optimizer and the loss function described in the previous paragraph.

## Statistical analysis

To compare the output of the model between conditions, we first obtained a distribution of values across several independent realizations of the model. Afterwards, we compared the overlap in the resulting distributions by means of Cohen's d, a measure of effect size. We did not report the results in terms of p-values or other metrics that depend on the statistical power, since this can be increased artificially by computing additional realizations of the model.

## Acknowledgements

The authors thanks the Multi-Partner Consortium to Expand Dementia Research in Latin America (ReDLat) as well as the patients and their relatives. YSP is supported by European Union's Horizon 2020 research and innovation program under the Marie Sklodowska-Curie grant 896354. AI is partially supported by grants ANID/FONDECYT Regular (1210195 and 1210176 and 1220995); ANID/FONDAP/15150012; ANID/PIA/ANILLOS ACT210096; ANID/FONDEF ID20I10152 and ID22I10029; ANID/FONDAP 15150012; Takeda CW2680521 and the MULTI-PARTNER CONSORTIUM TO EXPAND DEMENTIA RESEARCH IN LATIN AMERICA [ReDLat], supported by National Institutes of Health, National Institutes of Aging (R01 AG057234), Alzheimer's Association (SG-20-725707), Rainwater

Charitable foundation – Tau Consortium, and Global Brain Health Institute. ET is supported by PICT-2019–02294 (Agencia I+D+i, Argentina) and ANID/FONDECYT Regular 1220995 (Chile). The content is solely the responsibility of the authors and does not represent the official views of these institutions.

## Additional information

### Funding

| Funder | Grant reference number | Author |
|---|---|---|
| Marie Skłodowska-Curie Actions | 896354 | Yonatan Sanz Perl |
| Fondo Nacional de Desarrollo Científico y Tecnológico | 1210195 | Agustin Ibanez |
| Fondo Nacional de Desarrollo Científico y Tecnológico | 1210176 | Agustin Ibanez |
| Fondo Nacional de Desarrollo Científico y Tecnológico | 1220995 | Agustin Ibanez Enzo Tagliazucchi |
| Fondo de Financiamiento de Centros de Investigación en Áreas Prioritarias | 15150012 | Agustin Ibanez |
| Comisión Nacional de Investigación Científica y Tecnológica | ACT210096 | Agustin Ibanez |
| Comisión Nacional de Investigación Científica y Tecnológica | ID20I10152 | Agustin Ibanez |
| Comisión Nacional de Investigación Científica y Tecnológica | ID22I10029 | Agustin Ibanez |
| Takeda | CW2680521 | Agustin Ibanez |
| National Institute on Aging | Multi-Partner Consortium to Expand Dementia Research in Latin America (ReDLat) R01 AG057234 | Agustin Ibanez |
| Alzheimer's Association | 20-725707 | Agustin Ibanez |
| Rainwater Charitable Foundation | | Agustin Ibanez |
| Agencia Nacional de Promoción de la Investigación, el Desarrollo Tecnológico y la Innovación | PICT-2019-02294 | Enzo Tagliazucchi |
| Tau Consortium | | Agustin Ibanez |
| Global Brain Health Institute | | Agustin Ibanez |

The funders had no role in study design, data collection and interpretation, or the decision to submit the work for publication.

### Author contributions

Yonatan Sanz Perl, Conceptualization, Software, Visualization, Methodology, Writing - original draft; Sol Fittipaldi, Cecilia Gonzalez Campo, Methodology; Sebastián Moguilner, Josephine Cruzat, Rubén

Herzog, Pavel Prado, Writing – review and editing; Matias E Fraile-Vazquez, Data curation; Morten L Kringelbach, Gustavo Deco, Methodology, Writing – review and editing; Agustin Ibanez, Conceptualization, Supervision, Writing - original draft; Enzo Tagliazucchi, Conceptualization, Supervision, Visualization, Writing - original draft

## Author ORCIDs
Yonatan Sanz Perl (ID) http://orcid.org/0000-0002-1270-5564
Gustavo Deco (ID) http://orcid.org/0000-0002-8995-7583
Pavel Prado (ID) http://orcid.org/0000-0002-1324-6400
Agustin Ibanez (ID) http://orcid.org/0000-0001-6758-5101

## Ethics
All participants provided written informed consent pursuant to the Declaration of Helsinki. The study was approved by the Ethics Committees of the involved institutions (Comite de Ética Científico Servicio de Salud Metropolitano Oriente, IRB00007701; Instituto Alberto C. Taquini de Investigaciones en Medicina Traslacional, IRB00013030; Universidad Adolfo Ibañez, IRB00012394).

## Decision letter and Author response
Decision letter https://doi.org/10.7554/eLife.83970.sa1
Author response https://doi.org/10.7554/eLife.83970.sa2

## Additional files

### Supplementary files
• Supplementary file 1. The demographic information of the participants (CNT, AD, and FTD): gender, education level, age, and the results of cognitive assessments.

• Supplementary file 2. Gray matter atrophy for AD and FTD. (a) Gray matter atrophy areas of participants with AD (p<0.001, FWE-cluster-corrected for multiple comparisons). (b) Gray matter atrophy areas of participants with FTD (p<0.001, FWE-cluster-corrected for multiple comparisons).

• MDAR checklist

### Data availability
Data and codes required to reproduce the findings of this manuscript are freely available at the following OSF webpage: https://tinyurl.com/27652jkz.

The following dataset was generated:

| Author(s) | Year | Dataset title | Dataset URL | Database and Identifier |
|---|---|---|---|---|
| Tagliazucchi E | 2022 | Model-based whole-brain perturbational landscape of neurodegenerative diseases | https://osf.io/wunzt/ | Open Science Framework, wunzt |

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
