## [Editor Report]

Sanz Perl and colleagues provide important insights regarding the application of computational brain models from neurodegenerative diseases to evaluate brain stimulation protocols in silico. Solid evidence is provided for the disease-specificity of the framework, however, the real-world impact of such stimulation protocols to alleviate psychiatric and neurological symptoms remains to be evaluated.

---

## [Decision Letter]

**Decision letter after peer review:**

Thank you for submitting your article "Model-based whole-brain perturbational landscape of neurodegenerative diseases" for consideration by *eLife*. Your article has been reviewed by 2 peer reviewers, and the evaluation has been overseen by a Reviewing Editor and Jeannie Chin as the Senior Editor. The following individual involved in the review of your submission has agreed to reveal their identity: Jordi A Matias-Guiu (Reviewer #2).

Essential revisions:

1. In general, I think the methodology (especially the VAE) should be explained more substantially in the Results, to make the paper easier to follow in the order in which it is presented.

2. Further information is required from participants. For example, age, years of education, years since symptom onset, some cognitive tests, etc. A table reporting this information may be of interest. Further, patients and controls are not "matched", because there is no matching in this study (e.g. 1:1). This sentence that participants are matched in terms of age should be amended.

3. "All patients with bvFTD were in the mild stage of the disease and presented frontal compromise". What is the meaning of "frontal compromise"? Frontal syndrome or frontal atrophy in MRI?

4. It seems that AD did not have amyloid or tau biomarkers. This should be added as a limitation.

5. I wonder if the authors could elaborate on why wave stimulation simulates tACS and synch/noise resembles tDCS? That is stated, but not actually explained, in the manuscript.

6. It would be helpful to assess the robustness of results against alternative atlases.

7. The claim of disease-specificity is weakened by the use of a combined atrophy map to fit both conditions: it would be important to show the results of using the separate AD and FTD maps to fit AD and FTD, respectively (and the reverse): this would be a strong test for specificity, and it would bolster the case for disease-specific applications.

8. How does distance from controls in the 2D latent space match with GOF to the controls' FC? Are the two measures correlated? This would be important to identify how much the VAE really adds to the workflow.

9. Figure 2: I apologise if I missed this, but what are the effect sizes being computed against?

10. Regional results: I appreciate the authors' choice to not use p-values due to their limited meaning in the context of computational models that can be overpowered. However, presumably, there is some likelihood of obtaining effect sizes greater than a given threshold, just by chance – and this should be corrected for, akin to the traditional correction for multiple comparisons.

11. For Figure 5, it would be very helpful to show, not just the single points, but rather the area occupied by each condition, to facilitate the assessment of whether a given stimulation is successful, or simply falls more or less short of the target.

12. It would be a powerful demonstration if any of the results could be related to behavioural aspects, to show that the results are not confined to neuroimaging alone – after all, behavioural effects are the goal for actual brain stimulation.

13. Although the paper is mainly based on bvFTD, in terms of non-invasive brain stimulation, there are some references that may be of interest to support the rationale of the study. For instance, the works by Tsapkini (10.1007/s10072-019-04229-z, 10.1016/j.bandl.2019.104707, 10.1016/j.trci.2018.08.002) or about personalized TMS in PPA (10.3233/JAD-210566).

---

## [Author Response]

Essential revisions:1. In general, I think the methodology (especially the VAE) should be explained more substantially in the Results, to make the paper easier to follow in the order in which it is presented.

Thanks for this suggestion. We have included an additional overview of the methodology in the Results section of the revised version of the manuscript.

In the first paragraph of the Results:

“Briefly, VAEs are deep neural networks with autoencoder (AE) architecture (Kingma and Welling, 2013), which are trained to map inputs to probability distributions in latent space by minimizing the error between the input and the output. This output corresponds to the input as reproduced from the latent space values. Moreover, VAEs can be regularized during the training process to produce meaningful outputs after the decoding step, as well as to ensure continuity between the outputs and the corresponding choice of latent space values. The most common architecture of this network can be subdivided into three parts: the encoder network, the middle variational layer (with units corresponding to the latent space), and decoder network. The encoder transforms the input into the latent space, which is typically of much lower dimension than the input and output layers. On the other hand, the decoder converts the values of the units in the latent space to the output space”.

2. Further information is required from participants. For example, age, years of education, years since symptom onset, some cognitive tests, etc. A table reporting this information may be of interest. Further, patients and controls are not "matched", because there is no matching in this study (e.g. 1:1). This sentence that participants are matched in terms of age should be amended.

We thank the Reviewer for this comment. Indeed, even though participants were not matched in a strict sense, the selected groups did not present significant differences in the age of the participants. Accordingly, we have changed the sentence mentioned by the Reviewer by the following:

“…no significant differences in the age of participants were observed between groups (Kruskal-Wallis test, p>0.05).”

Also, as requested by the Reviewer, we have included a new table with gender, education level, age, and the results of cognitive assessments (supplementary file 1).

3. "All patients with bvFTD were in the mild stage of the disease and presented frontal compromise". What is the meaning of "frontal compromise"? Frontal syndrome or frontal atrophy in MRI?

To clarify this point, as well as to show that the pathological groups presented expected patterns of brain atrophy according to their classification as AD or bvFTD patients, we have added additional supplementary data displaying the atrophy maps of AD and bvFTD compared to CNT. Also, because its meaning was unclear, we have also deleted the sentence “All patients with bvFTD were in the mild stage of the disease and presented frontal compromise” from the revised version of the manuscript, and replaced it by the following:

“54 participants fulfilled the NINCDS-ADRDA criteria for typical Alzheimer’s Disease (AD) (Dubois et al., 2007), presented memory deficits, and showed atrophy in middle-temporal, hippocampal and posterior regions, among others (Figure 2 —figure supplement 1 in the Supplementary material). 31 participants met the revised criteria for probable behavioral variant frontotemporal dementia (bvFTD) (Piguet et al., 2011), presented behavioral and social deficits according to caregivers, and showed fronto-temporo-insular atrophy (see Figure 2 —figure supplement 1 and detailed explanation on the atrophy maps computation in the Supplementary material).”

We have also included as Supplementary material a detailed explanation on how the atrophy maps were computed and the tables included as supplementary file 2a and 2b.

The atrophy pattern of participants with AD and bvFTD was calculated by comparing their grey matter W-maps with those of controls using two-sample t-tests in SPM12. The α level was set at p <.001, cluster-corrected for multiple comparisons. Localization was derived from the AAL atlas.

4. It seems that AD did not have amyloid or tau biomarkers. This should be added as a limitation.

We Thank the Reviewer for the comment. We have mentioned this as a limitation in the revised version of our manuscript.

“Another limitation of this work is related to the diagnosis of the patients, which was based on current clinical criteria. Mainstream dementia diagnosis may also include biomarkers such as deposition of amyloid-β and tau proteins. These are quantified through Positron Emission Tomography (PET) or plasma, especially in AD (Agrawal and Biswas, 2015). However, it is currently common for dementia research to use the standard clinical criteria as reported in this paper (Birba et al., 2022; Herzog et al., 2022; Legaz et al., 2022; Maito et al., 2023; Parra et al., 2022). Moreover, PET/plasma biomarkers present caveats for global settings. PET is not widely available (i.e., the percentage of PET access for patients in global settings is less than 1% (Parra et al., 2018, 2022)) nor cost-effective (Horgan et al., 2020; Parra et al., 2022). It does not provide a conclusive diagnosis (Brown et al., 2014) nor discriminates well between FTD variants (Ntymenou et al., 2021). Fluid biomarkers (i.e., plasma) are very promising, but are not yet massively accessible. More importantly, plasma biomarkers currently lack systematic validation in diverse and non-stereotypical populations (Parra et al., 2022) like the current sample. However, future works should (a) combine clinical and biomarkers criteria to model whole brain dynamics; and (b) use metabolic maps of tau and amyloid maps as priors to improve the fit of the whole brain modelling”.

5. I wonder if the authors could elaborate on why wave stimulation simulates tACS and synch/noise resembles tDCS? That is stated, but not actually explained, in the manuscript.

We appreciate this comment. We have elaborated an explanation of the rationale of the parallelism between the proposed in silico interventions and the actual tACS and tDCS stimulations. The following paragraph was added to the revised version the manuscript:

“We explored different forms of in silico external perturbations that emulate specific empirical perturbation protocols. Wave stimulation (periodic perturbation delivered at the dominant frequency of the local BOLD fluctuations) imitates the characteristics of tACS, considering that both approaches are based on an external periodical driver applied to the brain. The specific application of the nodal natural oscillatory frequency is based on reports that suggest electrophysiological oscillations can be synchronized by in-phase tACS stimulation (Helfrich et al., 2014), even though this mechanism has been recently disputed (Lafon et al., 2017) and further research is required for its validation. On the other hand, the simulated Sync/Noise stimulation increases/decreases the overall value of the bifurcation parameter underlying the switching of the dynamical regime of a specific brain region. This can be associated with a direct alteration in nodal neural excitability, which resembles the impact created by tDCS stimulation (Nitsche and Paulus, 2000).”

6. It would be helpful to assess the robustness of results against alternative atlases.

We thank the Reviewer for this observation and the opportunity to explain our rationale for atlas selection. The AAL atlas has been extensively employed in brain network research in dementia and AD (Sanz-Arigita et al., 2010; Seo, Lee, Lee, Park, Sohn, Choe, et al., 2013; Seo, Lee, Lee, Park, Sohn, Lee, et al., 2013; Xiang et al., 2013) and in particular in FTD (Agosta et al., 2013; Filippi et al., 2013; Reyes et al., 2018; Sedeno et al., 2017; Sedeño et al., 2016; Whitwell et al., 2011). In consequence, to facilitate the comparison of our results with those of previous studies, we have chosen an atlas that has been systematically utilized in previous research on dementia.

Also, there are other reasons that support our choice. As there is no ground-truth parcellation of the cerebral cortex, its selection relies on (I) reproducibility, (II) clustering validity metrics, (III) multi-modal comparisons, and (IV) network analysis (Arslan et al., 2018). According to the previous findings (Arslan et al., 2018), no particular parcellation is consistently better across all assessment measures. Moreover, AAL performed similarly to other functional parcellation methods in this study (Arslan et al., 2018) although the subjects utilized were young (Asim et al., 2018; Bachli et al., 2020; McMillan et al., 2014; Park et al., 2022). However, there is no record of studies comparing brain parcellation in elderly and dementia samples.

Similarly, numerous studies have used the AAL parcellation for computational approaches in dementia samples (Asim et al., 2018; Bachli et al., 2020; Castellazzi et al., 2020; Moguilner et al., 2021; Sedeno et al., 2017). In that direction, a recent work from a subset of co-authors of this manuscript (Gonzalez-Gomez et al., 2023) empirically tested the impact of the parcellation in the performance of machine learning algorithms applied to dementia neuroimaging data. This analysis used a subsample of the data (structural graph connectivity and structural graph multifeature variables) to compare the performances when using different parcellations (i.e, AAL-116 and the human connectome project atlas). In both cases, the difference in performance was not statistically significant (p > 0.05) as assessed with nonparametric tests between the ROC curves (Venkatraman, 2000). Moreover, the most important features of the new atlas were consistent with the results obtained with the AAL atlas (i.e., features from parietal, occipital, and frontal areas).

Taken together, these reasons and results support the selection of the AAL atlas as a suitable parcellation for our study. We have also included a discussion of the potential limitations of using the AAL atlas:

“Finally, despite our choice of the AAL parcellation being based on its widespread adoption in dementia research (Agosta et al., 2013; Lord et al., 2016; Reyes et al., 2018; Sedeno et al., 2017; Whitwell et al., 2011), future research may require a systematic exploration of other brain parcellations applied to the dementia population. In that direction, to compare the differences across parcellations, a recent study by Gonzalez et al. used the AAL and HCP atlas (Glasser and Essen, 2011) parcellations on a dementia sample (graph connectivity and graph multifeature in both modalities) and did not find significant differences (Gonzalez-Gomez et al., 2023)”.

7. The claim of disease-specificity is weakened by the use of a combined atrophy map to fit both conditions: it would be important to show the results of using the separate AD and FTD maps to fit AD and FTD, respectively (and the reverse): this would be a strong test for specificity, and it would bolster the case for disease-specific applications.

We agree with this observation. The rationale behind the use of a combined atrophy map to fit both conditions is based on the fact that both maps are highly correlated R=0.75, p<0.001 (as it is also possible to observe in the brain renders presented in Figure R1 and in the scatter plots in Figure R2, left panel). At the same time, there are no significant correlations between the atrophy maps of AD and FTD with the PD atrophy map (PD vs. Ad; R=-0.03, p=0.76; PD vs. FTD; R=-0.22, p=0.05, see Figure R3). Therefore, since the atrophy maps of both conditions were similar, we decided to combine them to obtained improved and less noisy atrophy estimates.

Nevertheless, following the Reviewer's suggestion, we computed the results of Figure 2 using the separate AD and FTD maps to fit each group of participants. As shown in Figure 2—figure supplement 3, the model using the FTD atrophy map showed the best fit to the FTD fMRI empirical data, in line with the claim of disease-specificity. In the case of the AD model, both atrophy maps (FTD and AD) gave equally good results. We have included these observations in the revised main text and as new figures in the supplementary material (Figure 2 —figure supplement 2 and 3).

8. How does distance from controls in the 2D latent space match with GOF to the controls' FC? Are the two measures correlated? This would be important to identify how much the VAE really adds to the workflow.

We welcome the Reviewer’s comment. First, it is important to note that the use of the VAE adds information to the workflow regarding the 2D perturbational landscape, e.g., the particular trajectory of each perturbation. Therefore, regardless of the correlation between the distance to CNT and the GOF, the 2D latent space embedding provides information that goes beyond a single numerical index. In this direction, one-dimensional measures comparing perturbations with the target state (controls in this case), such as the GOF, are strictly less informative than the information provided by the 2D VAE latent space.

To clarify this point, we systematically decoded points within a 20x20 grid in the latent space and computed 1-GOF between the decoded FCs and the controls’ FC. In Figure 5—figure supplement 1, we represent the 1-GOF map together with the controls’ centre (blue circle), AD+ centre (red circle), and the minimal distance of all the possible perturbations of the AD+ condition (following the wave perturbative approach, black triangles). It can be seen that the 1-GOF respect to the CNT is minimal near the centre of the CNT group (as indicated by dark blue colour). However, 1-GOF did not increase homogeneously in all directions of the latent space. This complicated structure would be difficult to capture using a single numerical index, and adds support to the use of the 2D VAE latent space to understand how the simulated perturbation changes whole brain FC relative to the typical FC corresponding to the different groups of participants.

9. Figure 2: I apologise if I missed this, but what are the effect sizes being computed against?

We computed the effect size against the prior with the best goodness of fit obtained for each group, indicated with dashed line in each subpanel, e.g., in the fit of healthy group the comparison was performed against the RSN prior, which gave the best goodness of fit values in that particular case. We have clarified this in the revised version of the manuscript.

10. Regional results: I appreciate the authors' choice to not use p-values due to their limited meaning in the context of computational models that can be overpowered. However, presumably, there is some likelihood of obtaining effect sizes greater than a given threshold, just by chance – and this should be corrected for, akin to the traditional correction for multiple comparisons.

The correction for multiple comparisons is necessary when several statistical tests are applied to investigate the presence of significant results, evaluated as the probability of observing the measured data under the assumption of the null hypothesis. Since the outcome of each individual test is interpreted in probabilistic terms, performing a large number of tests increases the likelihood of observing results below the standard significance threshold due to statistical fluctuations, thus requiring a correction.

In the case of effect size computed using the Cohen’s D, we assess the extent to which two distributions differ in their means relative to their variance. These distributions are obtained by collecting the outcomes of a large number of simulations, and are reproducible, i.e. repeating the simulations results in very similar distributions of values (Ipiña et al., 2020). The lack of interpretation in probabilistic terms together with the robustness of the distributions across simulations ensures that large Cohen D values do not arise as spurious results due to statistical fluctuations (i.e., as opposed to the case of false positives in frequentist hypothesis testing, if the numerical experiments were repeated, the Cohen D values would remain similar – in particular those deemed as indicative of large effect size). Note that the use of frequentist hypothesis testing to assess the significance of the outcome of computer simulations is problematic since an arbitrary high number of replications can be performed, therefore it has been suggested to focus instead on effect sizes (White et al., 2014).

11. For Figure 5, it would be very helpful to show, not just the single points, but rather the area occupied by each condition, to facilitate the assessment of whether a given stimulation is successful, or simply falls more or less short of the target.

Thanks for this comment. We have included the area occupied by each condition in Figure 5 following the suggestion. The figure was improved as follows:

12. It would be a powerful demonstration if any of the results could be related to behavioural aspects, to show that the results are not confined to neuroimaging alone – after all, behavioural effects are the goal for actual brain stimulation.

This is an excellent proposal; however, it represents one step further in the whole-brain modelling framework. We have proposed a combined framework where modelling was used to obtain sufficient samples to train a VAE. However, as in previous works (Deco et al., 2017; Ipiña et al., 2020; Jobst et al., 2017) ,the whole-brain model was introduced at the group level, which alleviates potential issues associated with noisy measurements and individual variance. To correlate the results with behavioural outcomes of single patients, we should extend our framework to model individual brains and their dynamics, which represents a natural way forward. Future work will aim to provide new tools to diagnose and predict the effects of different manipulations in patients at the individual level, and to test the results by comparing them with behavioural variables. In the revised version of our manuscript, we have acknowledged this as a limitation of the current work and we proposed a way forward for future research:

“We conducted group-level analyses, which constitute a valuable proof of principle for applying whole-brain modeling to neurodegenerative diseases, but fall short of providing information at the single subject level, which would be required to correlate the results of the model with behavioral outcomes, especially given results showing that the therapeutic effect of transcranial stimulation depends on individual characteristics of brain anatomy (Unal et al. 2020; Tsapkini et al. 2018; de Aguiar et al. 2020; Pytel et al. 2021). For this purpose, models tailored to data from individual patients will be required (Arianna Menardi 2022).”

13. Although the paper is mainly based on bvFTD, in terms of non-invasive brain stimulation, there are some references that may be of interest to support the rationale of the study. For instance, the works by Tsapkini (10.1007/s10072-019-04229-z, 10.1016/j.bandl.2019.104707, 10.1016/j.trci.2018.08.002) or about personalized TMS in PPA (10.3233/JAD-210566).

We have included these interesting references provided by the Reviewer, and we have also included other references in the same direction:

Pini, L., Pizzini, F. B., Boscolo-Galazzo, I., Ferrari, C., Galluzzi, S., Cotelli, M., … and Pievani, M. (2022). Brain network modulation in Alzheimer's and frontotemporal dementia with transcranial electrical stimulation. *Neurobiology of Aging*, *111*, 24-34.

Birba, A., Ibáñez, A., Sedeño, L., Ferrari, J., García, A. M., and Zimerman, M. (2017). Non-invasive brain stimulation: a new strategy in mild cognitive impairment?. *Frontiers in Aging Neuroscience*, 16.

References

Agosta, F., Sala, S., Valsasina, P., Meani, A., Canu, E., Magnani, G., Cappa, S. F., Scola, E., Quatto, P., Horsfield, M. A., and others. (2013). Brain network connectivity assessed using graph theory in frontotemporal dementia. *Neurology*, *81*(2), 134–143.

Agrawal, M., and Biswas, A. (2015). Molecular diagnostics of neurodegenerative disorders. *Frontiers in Molecular Biosciences*, *2*(SEP), 54. https://doi.org/10.3389/FMOLB.2015.00054/BIBTEX

Arslan, S., Ktena, S. I., Makropoulos, A., Robinson, E. C., Rueckert, D., and Parisot, S. (2018). Human brain mapping: A systematic comparison of parcellation methods for the human cerebral cortex. *Neuroimage*, *170*, 5–30.

Asim, Y., Raza, B., Malik, A. K., Rathore, S., Hussain, L., and Iftikhar, M. A. (2018). A multi-modal, multi-atlas-based approach for Alzheimer detection via machine learning. *International Journal of Imaging Systems and Technology*, *28*(2), 113–123.

Bachli, M. B., Sedeño, L., Ochab, J. K., Piguet, O., Kumfor, F., Reyes, P., Torralva, T., Roca, M., Cardona, J. F., Campo, C. G., and others. (2020). Evaluating the reliability of neurocognitive biomarkers of neurodegenerative diseases across countries: a machine learning approach. *Neuroimage*, *208*, 116456.

Birba, A., Santamaría-García, H., Prado, P., Cruzat, J., Ballesteros, A. S., Legaz, A., Fittipaldi, S., Duran-Aniotz, C., Slachevsky, A., Santibañez, R., Sigman, M., García, A. M., Whelan, R., Moguilner, S., and Ibáñez, A. (2022). Allostatic-Interoceptive Overload in Frontotemporal Dementia. *Biological Psychiatry*, *92*(1), 54–67. https://doi.org/10.1016/J.BIOPSYCH.2022.02.955

Brown, R. K. J., Bohnen, N. I., Wong, K. K., Minoshima, S., and Frey, K. A. (2014). Brain PET in Suspected Dementia: Patterns of Altered FDG Metabolism. *Https://Doi.Org/10.1148/Rg.343135065*, *34*(3), 684–701. https://doi.org/10.1148/RG.343135065

Castellazzi, G., Cuzzoni, M. G., Cotta Ramusino, M., Martinelli, D., Denaro, F., Ricciardi, A., Vitali, P., Anzalone, N., Bernini, S., Palesi, F., and others. (2020). A machine learning approach for the differential diagnosis of alzheimer and vascular dementia fed by MRI selected features. *Frontiers in Neuroinformatics*, 25.

Deco, G., Kringelbach, M. L., Jirsa, V. K., and Ritter, P. (2017). The dynamics of resting fluctuations in the brain: metastability and its dynamical cortical core. *Sci. Rep.*, *7*(1), 3095. https://doi.org/10.1038/s41598-017-03073-5

Dubois, B., Feldman, H. H., Jacova, C., DeKosky, S. T., Barberger-Gateau, P., Cummings, J., Delacourte, A., Galasko, D., Gauthier, S., Jicha, G., Meguro, K., O’Brien, J., Pasquier, F., Robert, P., Rossor, M., Salloway, S., Stern, Y., Visser, P. J., and Scheltens, P. (2007). Research criteria for the diagnosis of Alzheimer’s disease: revising the NINCDS–ADRDA criteria. *The Lancet Neurology*, *6*(8), 734–746. https://doi.org/10.1016/S1474-4422(07)70178-3

Filippi, M., Agosta, F., Scola, E., Canu, E., Magnani, G., Marcone, A., Valsasina, P., Caso, F., Copetti, M., Comi, G., and others. (2013). Functional network connectivity in the behavioral variant of frontotemporal dementia. *Cortex*, *49*(9), 2389–2401.

Glasser, M. F., and Essen, D. C. Van. (2011). Mapping Human Cortical Areas in vivo Based on Myelin Content as Revealed by T1- and T2-Weighted MRI. *Journal of Neuroscience*, *31*(32), 11597–11616. https://doi.org/10.1523/JNEUROSCI.2180-11.2011

Gonzalez-Gomez, R., Ibañez, A., Moguilner, S., and Rubinov, M. (2023). Multiclass characterization of frontotemporal dementia variants via multimodal brain network computational inference. *Network Neuroscience*, *7*(1), 322–350. https://doi.org/10.1162/NETN_A_00285

Helfrich, R. F., Schneider, T. R., Rach, S., Trautmann-Lengsfeld, S. A., Engel, A. K., and Herrmann, C. S. (2014). Entrainment of Brain Oscillations by Transcranial Alternating Current Stimulation. *Current Biology*, *24*(3), 333–339. https://doi.org/10.1016/J.CUB.2013.12.041

Herzog, R., Rosas, F. E., Whelan, R., Fittipaldi, S., Santamaria-Garcia, H., Cruzat, J., Birba, A., Moguilner, S., Tagliazucchi, E., Prado, P., and Ibanez, A. (2022). Genuine high-order interactions in brain networks and neurodegeneration. *Neurobiology of Disease*, *175*, 105918. https://doi.org/10.1016/J.NBD.2022.105918

Horgan, D., Nobili, F., Teunissen, C., Grimmer, T., Mitrecic, D., Ris, L., Pirtosek, Z., Bernini, C., Federico, A., Blackburn, D., Logroscino, G., and Scarmeas, N. (2020). Biomarker Testing: Piercing the Fog of Alzheimer’s and Related Dementia. *Biomedicine Hub*, *5*(3), 1–22. https://doi.org/10.1159/000511233

Ipiña, I. P., Kehoe, P. D., Kringelbach, M., Laufs, H., Ibañez, A., Deco, G., Perl, Y. S., and Tagliazucchi, E. (2020). Modeling regional changes in dynamic stability during sleep and wakefulness. *NeuroImage*, *215*, 116833. https://doi.org/10.1016/J.NEUROIMAGE.2020.116833

Jobst, B. M., Hindriks, R., Laufs, H., Tagliazucchi, E., Hahn, G., Ponce-Alvarez, A., Stevner, A. B. A., Kringelbach, M. L., and Deco, G. (2017). Increased Stability and Breakdown of Brain Effective Connectivity During Slow-Wave Sleep: Mechanistic Insights from Whole-Brain Computational Modelling. *Scientific Reports 2017 7:1*, *7*(1), 1–16. https://doi.org/10.1038/s41598-017-04522-x

Kingma, D. P., and Welling, M. (2013). Auto-Encoding Variational Bayes. *2nd International Conference on Learning Representations, ICLR 2014 – Conference Track Proceedings*. https://arxiv.org/abs/1312.6114v10

Lafon, B., Henin, S., Huang, Y., Friedman, D., Melloni, L., Thesen, T., Doyle, W., Buzsáki, G., Devinsky, O., Parra, L. C., and Liu, A. A. (2017). Low frequency transcranial electrical stimulation does not entrain sleep rhythms measured by human intracranial recordings. *Nature Communications 2017 8:1*, *8*(1), 1–14. https://doi.org/10.1038/s41467-017-01045-x

Legaz, A., Abrevaya, S., Dottori, M., González Campo, C., Birba, A., Martorell Caro, M., Aguirre, J., Slachevsky, A., Aranguiz, R., Serrano, C., Gillan, C. M., Leroi, I., García, A. M., Fittipaldi, S., and Ibañez, A. (2022). Multimodal mechanisms of human socially reinforced learning across neurodegenerative diseases. *Brain*, *145*(3), 1052–1068. https://doi.org/10.1093/BRAIN/AWAB345

Lord, A., Ehrlich, S., Borchardt, V., Geisler, D., Seidel, M., Huber, S., Murr, J., and Walter, M. (2016). Brain parcellation choice affects disease-related topology differences increasingly from global to local network levels. *Psychiatry Research: Neuroimaging*, *249*, 12–19.

Maito, M. A., Santamaría-García, H., Moguilner, S., Possin, K. L., Godoy, M. E., Avila-Funes, J. A., Behrens, M. I., Brusco, I. L., Bruno, M. A., Cardona, J. F., Custodio, N., García, A. M., Javandel, S., Lopera, F., Matallana, D. L., Miller, B., Okada de Oliveira, M., Pina-Escudero, S. D., Slachevsky, A., … Ibañez, A. (2023). Classification of Alzheimer’s disease and frontotemporal dementia using routine clinical and cognitive measures across multicentric underrepresented samples: a cross sectional observational study. *The Lancet Regional Health – Americas*, *17*, 100387. https://doi.org/10.1016/J.LANA.2022.100387

McMillan, C. T., Avants, B. B., Cook, P., Ungar, L., Trojanowski, J. Q., and Grossman, M. (2014). The power of neuroimaging biomarkers for screening frontotemporal dementia. *Human Brain Mapping*, *35*(9), 4827–4840.

Moguilner, S., García, A. M., Perl, Y. S., Tagliazucchi, E., Piguet, O., Kumfor, F., Reyes, P., Matallana, D., Sedeño, L., and Ibáñez, A. (2021). Dynamic brain fluctuations outperform connectivity measures and mirror pathophysiological profiles across dementia subtypes: A multicenter study. *NeuroImage*, *225*, 117522. https://doi.org/10.1016/J.NEUROIMAGE.2020.117522

Nitsche, M. A., and Paulus, W. (2000). Excitability changes induced in the human motor cortex by weak transcranial direct current stimulation. *The Journal of Physiology*, *527*(Pt 3), 633. https://doi.org/10.1111/J.1469-7793.2000.T01-1-00633.X

Ntymenou, S., Tsantzali, I., Kalamatianos, T., Voumvourakis, K. I., Kapaki, E., Tsivgoulis, G., Stranjalis, G., and Paraskevas, G. P. (2021). Blood Biomarkers in Frontotemporal Dementia: Review and Meta-Analysis. *Brain Sciences 2021, Vol. 11, Page 244*, *11*(2), 244. https://doi.org/10.3390/BRAINSCI11020244

Park, B., Choi, B. J., Lee, H., Jang, J.-H., Roh, H. W., Kim, E. Y., Hong, C. H., Son, S. J., and Yoon, D. (2022). Modeling brain volume using deep learning-based physical activity features in patients with dementia. *Frontiers in Neuroinformatics*, *16*.

Parra, M. A., Baez, S., Allegri, R., Nitrini, R., Lopera, F., Slachevsky, A., Custodio, N., Lira, D., Piguet, O., Kumfor, F., Huepe, D., Cogram, P., Bak, T., Manes, F., and Ibanez, A. (2018). Dementia in Latin America. *Neurology*, *90*(5), 222–231. https://doi.org/10.1212/WNL.0000000000004897

Parra, M. A., Orellana, P., Leon, T., Victoria, C. G., Henriquez, F., Gomez, R., Avalos, C., Damian, A., Slachevsky, A., Ibañez, A., Zetterberg, H., Tijms, B. M., Yokoyama, J. S., Piña-Escudero, S. D., Cochran, J. N., Matallana, D. L., Acosta, D., Allegri, R., Arias-Suárez, B. P., … Duran-Aniotz, C. (2022). Biomarkers for dementia in Latin American countries: Gaps and opportunities. *Alzheimer’s and Dementia*, *6*, 21. https://doi.org/10.1002/ALZ.12757

Piguet, O., Hornberger, M., Mioshi, E., and Hodges, J. R. (2011). Behavioural-variant frontotemporal dementia: diagnosis, clinical staging, and management. *The Lancet Neurology*, *10*(2), 162–172.

Reyes, P., Ortega-Merchan, M. P., Rueda, A., Uriza, F., Santamaria-Garc\’\ia, H., Rojas-Serrano, N., Rodriguez-Santos, J., Velasco-Leon, M. C., Rodriguez-Parra, J. D., Mora-Diaz, D. E., and others. (2018). Functional connectivity changes in behavioral, semantic, and nonfluent variants of frontotemporal dementia. *Behavioural Neurology*, *2018*.

Sanz-Arigita, E. J., Schoonheim, M. M., Damoiseaux, J. S., Rombouts, S. A. R. B., Maris, E., Barkhof, F., Scheltens, P., and Stam, C. J. (2010). Loss of ‘small-world’networks in Alzheimer’s disease: graph analysis of FMRI resting-state functional connectivity. *PloS One*, *5*(11), e13788.

Sedeño, L., Couto, B., Garc\’\ia-Cordero, I., Melloni, M., Baez, S., Sepúlveda, J. P. M., Fraiman, D., Huepe, D., Hurtado, E., Matallana, D., and others. (2016). Brain network organization and social executive performance in frontotemporal dementia. *Journal of the International Neuropsychological Society*, *22*(2), 250–262.

Sedeno, L., Piguet, O., Abrevaya, S., Desmaras, H., Garc\’\ia-Cordero, I., Baez, S., de la Fuente, L., Reyes, P., Tu, S., Moguilner, S., and others. (2017). Tackling variability: A multicenter study to provide a gold-standard network approach for frontotemporal dementia. *Human Brain Mapping*, *38*(8), 3804–3822.

Seo, E. H., Lee, D. Y., Lee, J.-M., Park, J.-S., Sohn, B. K., Choe, Y. M., Byun, M. S., Choi, H. J., and Woo, J. I. (2013). Influence of APOE genotype on whole-brain functional networks in cognitively normal elderly. *PloS One*, *8*(12), e83205.

Seo, E. H., Lee, D. Y., Lee, J.-M., Park, J.-S., Sohn, B. K., Lee, D. S., Choe, Y. M., and Woo, J. I. (2013). Whole-brain functional networks in cognitively normal, mild cognitive impairment, and Alzheimer’s disease. *PloS One*, *8*(1), e53922.

Venkatraman, E. S. (2000). A permutation test to compare receiver operating characteristic curves. *Biometrics*, *56*(4), 1134–1138.

White, J. W., Rassweiler, A., Samhouri, J. F., Stier, A. C., and White, C. (2014). Ecologists should not use statistical significance tests to interpret simulation model results. *Oikos*, *123*(4), 385–388. https://doi.org/10.1111/J.1600-0706.2013.01073.X

Whitwell, J. L., Josephs, K. A., Avula, R., Tosakulwong, N., Weigand, S. D., Senjem, M. L., Vemuri, P., Jones, D. T., Gunter, J. L., Baker, M., and others. (2011). Altered functional connectivity in asymptomatic MAPT subjects: a comparison to bvFTD. *Neurology*, *77*(9), 866–874.

Xiang, J., Guo, H., Cao, R., Liang, H., and Chen, J. (2013). An abnormal resting-state functional brain network indicates progression towards Alzheimer’s disease. *Neural Regeneration Research*, *8*(30), 2789.